# MDP Planning as Policy Inference

## Abstract

We formulate episodic Markov decision process (MDP) planning as Bayesian inference over *policies*. The primary contribution is conceptual: the policy itself is treated as the latent variable, and expected return defines an unnormalized posterior density over policies. This preserves the standard expected-return objective, in contrast to trajectory-centric planning-as-inference formulations that introduce auxiliary optimality variables and to entropy-regularized policy optimization methods that solve a different objective.

In the exact formulation, the posterior over deterministic policies induces what we define here as an *optimal stochastic policy under preference uncertainty*, namely the stochastic policy induced by that posterior. For discrete MDPs with stochastic transitions, we study variational sequential Monte Carlo (VSMC) as one approximate inference method for this posterior, introducing policy consistency under state revisitation and coupled transition randomness across particles.

Experiments on grid worlds, Blackjack, Triangle Tireworld, and Academic Advising examine the consequences of inference over policies and compare its induced behavior with entropy-regularized policy optimization. The results support the view that MDP planning can be naturally cast as Bayesian inference over policies.

## 1 Introduction

We cast episodic Markov decision process (MDP) planning as Bayesian inference over *policies*. The central claim of the paper is that the natural object of inference in planning is the policy itself: not a trajectory augmented with auxiliary optimality variables, and not a single stochastic policy trained under entropy regularization, but a posterior distribution over policies whose density is determined by expected return. Treating the policy as a latent variable has two benefits: it enables the use of general-purpose inference algorithms for planning, and it makes uncertainty about optimal behavior explicit as posterior dispersion rather than an artifact of approximation or heuristic regularization.

Casting planning and control as probabilistic inference is useful because it provides a principled modeling language for uncertainty and makes general-purpose approximate inference tools available for decision making. This perspective has a long history in probabilistic planning, control-as-inference, maximum-entropy reinforcement learning, and active inference (Dayan & Hinton, 1997; Toussaint & Storkey, 2006; Ziebart, 2010; Levine, 2018). Prior formulations typically modify the classical planning objective to fit a standard latent–observation template, for example by introducing entropy-regularized or evidence-based surrogate criteria. In these settings, stochasticity is often a modeling preference or an exploration device, and uncertainty over the solution of the original expected-return problem is not directly interpretable from the inferred policy.

This distinction in the object of inference matters. In trajectory-centric planning-as-inference methods, randomness is attached to latent trajectories and auxiliary optimality variables are introduced to recover a control objective. In entropy-regularized reinforcement learning, stochasticity is built directly into the optimized policy and reflects a modified objective. Here, by contrast, stochasticity arises from posterior uncertainty over deterministic policies, while the underlying objective remains the standard expected-return criterion.

In this work, we propose a Bayesian formulation that preserves the standard MDP objective. We define an unnormalized probability of optimality for each policy that is monotone in its expected return, yielding a posterior distribution whose modes coincide with return-maximizing policies, while posterior dispersion represents uncertainty over optimal behavior. Acting is performed by sampling from the posterior predictive distribution induced by this policy posterior. This yields what we define in Section 3.1 as the model-induced optimal stochastic policy under preference uncertainty, through a Thompson-sampling interpretation rather than through entropy regularization.

Once planning is formulated this way, approximate Bayesian inference is needed to work with the resulting posterior. In this paper we use variational sequential Monte Carlo (VSMC) (Naesseth et al., 2018; Maddison et al., 2017) as one approximate inference tool for discrete MDPs with stochastic transitions. The algorithmic development is therefore in service of the policy-inference formulation, rather than the main contribution of the paper. Concretely, our adaptation enforces policy consistency across revisited states and couples transition randomness across particles within a sweep so that particle weights reflect policy differences rather than independent realizations of simulator noise. We further interpret the reward scale as controlling uncertainty over agent preferences: larger rewards induce posterior concentration and near-deterministic behavior, while smaller rewards yield a diffuse posterior and a stochastic control policy that reflects preference uncertainty through the inferred variational posterior. This interpretation treats reward magnitudes as meaningful, not merely as an ordering over policies: a small return difference expresses weak preference between policies, whereas a large return difference expresses strong preference and therefore induces stronger posterior concentration.

**What this paper is and is not.** This paper is primarily about the *object of inference*. Our claim is that MDP planning can be cast naturally as Bayesian inference over *policies* while preserving the standard expected-return semantics. Accordingly, VSMC is used here as one approximate inference method for the resulting posterior; it is not the main conceptual contribution, nor do we present the method as a new entropy-regularized or "soft" planning objective. In contrast to trajectory-centric planning-as-inference methods, we do not introduce optimality variables or fictitious observations, and in contrast to entropy-regularized RL we do not optimize a single stochastic policy under an added entropy term.

Our contributions are:

- A formulation of episodic MDP planning as Bayesian inference over *policies*, in which the policy itself is the latent variable and expected return defines an unnormalized posterior density. In the exact formulation, this posterior induces what we define here as an **optimal stochastic policy under preference uncertainty**, namely, the stochastic policy obtained by marginalizing posterior uncertainty over deterministic policies, without introducing trajectory-level optimality variables or entropy regularization.

- An adaptation of VSMC for inference over deterministic policies in discrete MDPs with stochastic transitions, serving as an approximate inference algorithm for this policy posterior and including policy consistency under revisitation and coupled transition randomness across particles.

- An empirical evaluation of the *induced stochastic control policy* obtained by posterior predictive (Thompson-style) action sampling, and a comparison to discrete Soft Actor-Critic across diverse discrete benchmarks.

## 2 Background

### 2.1 Markov Decision Process

We consider episodic MDPs with state space $S$, action space $A$, stochastic transition kernel $\mathcal{T}(s' \mid s, a)$, reward function $R(s, a, s')$, initial state $s_1$, and a set of absorbing goal states $G \subseteq S$. A (Markov) policy $\pi$ maps states to actions; we write either $\pi(s) \in A$ for a deterministic policy or $p_\pi(a \mid s)$ for a stochastic policy. Episodes end upon first reaching a state in $G$. This absorbing-goal formulation is standard for episodic, goal-directed tasks, where interaction ends upon reaching a terminal state and policies are evaluated by

expected return up to termination (Sutton & Barto, 2018; Puterman, 1994). For planning and inference, we use a finite rollout horizon $H$ as an algorithm parameter and evaluate policies on truncated trajectories $\tau_\pi = (s_1, a_1, s_2, \ldots, a_H, s_{H+1})$ generated by iterating, for $t = 1, \ldots, H$,

$$s_{t+1} \leftarrow \text{STEP}(s_t, a_t), \qquad r_t \leftarrow \text{REWARD}(s_t, a_t, s_{t+1}), \tag{1}$$

where STEP samples $s_{t+1} \sim \mathcal{T}(\cdot \mid s_t, a_t)$ and REWARD computes $R(s_t, a_t, s_{t+1})$. If $s_t \in G$, the process remains in that absorbing state and accrues no further reward, so the truncated rollout continues only for notational convenience. If no goal state is reached by time $H$, the rollout is simply truncated. Throughout, inference algorithms access the MDP only through this simulator interface.

## 2.2 Variational Sequential Monte Carlo

Variational sequential Monte Carlo (VSMC) (Naesseth et al., 2018) is an algorithm for variational inference in models amenable to sequential Monte Carlo (SMC) (Gordon et al., 1993). In standard SMC, the posterior distribution of latent trajectories is approximated sequentially by $N$ weighted particles. At each step, particles representing partial latent trajectories are extended by sampling from a proposal kernel, assigned incremental importance weights, and optionally resampled according to those weights. An SMC sweep therefore produces two objects: a weighted particle approximation to the posterior, and an estimate of the marginal likelihood, or evidence. VSMC uses the latter object as a variational training signal: it treats the parametrized SMC proposal as a variational family and optimizes the expected SMC log-evidence estimate, with the expectation taken over proposal sampling and resampling in the SMC sweep.

Let $x_t \in \mathcal{X}_t$ be latent variables and $y_t \in \mathcal{Y}_t$ be observations, with $(\mathcal{X}_t, \mathcal{Y}_t)$ measurable spaces. Consider the state-space model

$$p(x_{1:H}, y_{1:H}) = p(x_1)p(y_1 \mid x_1) \prod_{t=1}^{H-1} p(x_{t+1} \mid x_t)p(y_{t+1} \mid x_{t+1}) \tag{2}$$

where $H$ is the horizon. The evidence is

$$Z = p(y_{1:H}). \tag{3}$$

Here $q_\lambda$ denotes a parameterized family of proposal kernels on the latent spaces, from which particles can be sampled sequentially and whose densities or masses can be evaluated for importance weighting. Given a proposal $q_\lambda$, SMC produces particles and weights $x_{1:t}^i, w_t^i\big|_{i=1}^N$ at each step $t$, and estimates the evidence by

$$\hat{Z}_N = \prod_{t=1}^{H} \frac{1}{N} \sum_{i=1}^{N} w_t^{(i)}. \tag{4}$$

The estimator $\hat{Z}_N$ is unbiased for $Z$, while $\log \hat{Z}_N$ is a biased estimate of $\log Z$ by Jensen's inequality. VSMC nevertheless uses the log estimate as a variational objective,

$$\mathcal{L}_{\text{VSMC}}(\lambda) = \mathbb{E}_{\text{SMC}(q_\lambda)} \left[ \log \hat{Z}_N \right], \tag{5}$$

where the expectation is over proposal sampling and resampling in the SMC sweep. The objective is a surrogate evidence lower bound for the marginal likelihood (Naesseth et al., 2018).

The proposal learned by VSMC should be understood as an approximation to a posterior conditional over the next latent variable, not merely as a locally adapted proposal depending only on the next observation. This is not a new result of the present paper; it is a reformulation of the variational interpretation of SMC developed by Naesseth et al. (2018). We state it explicitly here to fix notation and to make clear which proposal distribution is being approximated before specializing the construction to policy inference.

**Proposition 1** (Posterior-conditional target of the VSMC proposal)**.** *Let* $p(x_{1:H} \mid y_{1:H})$ *be the posterior over a latent trajectory. For fixed observations* $y_{1:H}$, *consider an autoregressive proposal family*

$$q_\lambda(x_{1:H} \mid y_{1:H}) = q_\lambda(x_1 \mid y_{1:H}) \prod_{t=1}^{H-1} q_\lambda(x_{t+1} \mid x_{1:t}, y_{1:H}). \tag{6}$$

*The maximizer of the variational evidence bound*

$$\mathcal{L}(\lambda) = \mathbb{E}_{q_\lambda(x_{1:H}|y_{1:H})} \left[ \log p(x_{1:H}, y_{1:H}) - \log q_\lambda(x_{1:H} \mid y_{1:H}) \right] \tag{7}$$

*satisfies, whenever the proposal family is expressive enough,*

$$q_\lambda^\star(x_1 \mid y_{1:H}) = p(x_1 \mid y_{1:H}) \tag{8}$$

*and, for each $t = 1, \ldots, H - 1$,*

$$q_\lambda^\star(x_{t+1} \mid x_{1:t}, y_{1:H}) = p(x_{t+1} \mid x_{1:t}, y_{1:H}). \tag{9}$$

*For the Markov state-space model above, this posterior conditional reduces to*

$$q_\lambda^\star(x_{t+1} \mid x_{1:t}, y_{1:H}) = p(x_{t+1} \mid x_t, y_{t+1:H}). \tag{10}$$

*Proof.* The variational bound can be written as

$$\mathcal{L}(\lambda) = \log p(y_{1:H}) - \mathrm{KL}\left( q_\lambda(x_{1:H} \mid y_{1:H}) \,\|\, p(x_{1:H} \mid y_{1:H}) \right). \tag{11}$$

The posterior factorizes by the chain rule as

$$p(x_{1:H} \mid y_{1:H}) = p(x_1 \mid y_{1:H}) \prod_{t=1}^{H-1} p(x_{t+1} \mid x_{1:t}, y_{1:H}). \tag{12}$$

Substituting this factorization and the factorization of $q_\lambda(x_{1:H} \mid y_{1:H})$ into the KL gives

$$\begin{aligned}
\mathrm{KL}\left( q_\lambda(x_{1:H} \mid y_{1:H}) \,\|\, p(x_{1:H} \mid y_{1:H}) \right) &= \mathrm{KL}\left( q_\lambda(x_1 \mid y_{1:H}) \,\|\, p(x_1 \mid y_{1:H}) \right) \\
&+ \sum_{t=1}^{H-1} \mathbb{E}_{q_\lambda(x_{1:t}|y_{1:H})} \left[ \mathrm{KL}\left( q_\lambda(\cdot \mid x_{1:t}, y_{1:H}) \,\|\, p(\cdot \mid x_{1:t}, y_{1:H}) \right) \right].
\end{aligned} \tag{13}$$

Each term is nonnegative. Hence the optimum is attained by setting each proposal factor equal to the corresponding posterior conditional.

It remains to simplify this conditional in the Markov case. Under the state-space factorization above, for $t = 1, \ldots, H - 1$,

$$p(x_{t+1} \mid x_{1:t}, y_{1:H}) = p(x_{t+1} \mid x_t, y_{t+1:H}), \tag{14}$$

because, conditional on $x_t$, the distribution of $x_{t+1}$ and the future observations $y_{t+1:H}$ is independent of the earlier latent states $x_{1:t-1}$ and observations $y_{1:t}$. This proves the final statement. $\square$

Thus, following the variational interpretation of SMC in Naesseth et al. (2018), the ideal proposal in a Markov model is the smoothing conditional distribution of the next latent state given the current latent state and the remaining observations. This differs from a myopic proposal that conditions only on the next observation. VSMC is therefore useful when the proposal can learn to predict latent choices that are favored by future evidence, a property used below when adapting VSMC to posterior inference over deterministic policies.

## 3 Probabilistic Model

Much of the control-as-inference literature introduces auxiliary *optimality* variables or other fictitious observations in order to cast planning into a standard latent–observed graphical model. Here we avoid such augmentation and instead work directly with an unnormalized target distribution over the object of interest—the policy. Posterior inference only requires access to an unnormalized density $\tilde{p}(\pi)$ (up to a multiplicative constant), possibly through an unbiased stochastic estimator of $\tilde{p}(\pi)$ (Andrieu & Roberts, 2009) or $\log \tilde{p}(\pi)$ (Hoffman et al., 2013).

Since we are interested in inferring a policy, the *policy* $\pi$ is the latent random variable. The objective of MDP planning is to identify policies that maximize expected return. To align the probabilistic model with this objective, we assign to each policy an unnormalized **probability of optimality** that is monotone in its expected return.

Specifically, we define the unnormalized log probability of a policy as the *expected return* obtained by the agent following the policy over a truncated rollout of length at most $H$, with expectation over trajectories $\tau_\pi$ distributed according to the dynamics induced by policy $\pi$:

$$\log \tilde{p}(\pi) = \mathbb{E}_{\tau_\pi} \sum_{t=1}^{H} R\left(s_t, a_t, s_{t+1}\right), \tag{15}$$

where $a_t = \pi(s_t)$ and $s_{t+1} \sim \mathcal{T}(\cdot \mid s_t, a_t)$ for $t = 1, \ldots, H$. Because goal states are absorbing, this sum coincides with the episode return when the goal is reached before $H$, and otherwise uses the rollout truncation horizon as a computational cutoff. This induces a Boltzmann–Gibbs distribution over *policies* (Ziebart et al., 2008; Todorov, 2006; Levine, 2018).

Note that neither actions nor states are treated as Bayesian random variables for which a posterior is sought. Although the policy (if stochastic policies are considered) induces a stochastic action selection rule and the environment induces stochastic state transitions, these are generative rather than inferential sources of randomness: actions and state transitions are sampled forward from their respective distributions rather than conditioned for the purpose of posterior inference. The randomness they induce propagates into the estimation of the unnormalized log probability of the policy, so that $\log \tilde{p}(\pi)$ is available only through noisy Monte Carlo evaluations—by computing the return of a single truncated rollout:

$$\log \widehat{\tilde{p}}(\pi) = \sum_{t=1}^{H} R\left(s_t, a_t, s_{t+1}\right). \tag{16}$$

Stochastic estimate (16) lays the basis for posterior inference of the policy distribution.

### 3.1 Posterior-Induced Stochastic Policy

The posterior over deterministic policies also induces a stochastic action-selection rule by marginalization. For any state $s$, define

$$p^\star(a \mid s) = \Pr_{\pi \sim p(\cdot)}[\pi(s) = a] = \mathbb{E}_{\pi \sim p(\cdot)}\left[\mathbf{1}\{\pi(s) = a\}\right], \tag{17}$$

where $p(\pi) \propto \tilde{p}(\pi)$ is the posterior over deterministic policies induced by Eq. (15). We refer to $p^\star(\cdot \mid s)$ as the **optimal stochastic policy under preference uncertainty**. This is a notion introduced in this paper: it denotes the stochastic policy obtained by marginalizing the posterior over deterministic policies under the model in Eq. (15). Its action probabilities are therefore the posterior probabilities that the corresponding actions are prescribed by posterior-supported deterministic policies at state $s$. Stochasticity is thus not introduced through entropy regularization; it arises from posterior uncertainty over which deterministic policy is preferred by the model.

### 3.2 Worked Tabular Examples

We use two small tabular examples to clarify the relationship between the policy-posterior construction, entropy-regularized control, and optimality-variable control-as-inference. The first example shows a setting in which the policy-posterior construction and entropy-regularized control coincide, while optimality-variable control-as-inference differs. The second example shows that the coincidence does not hold in general: when a state can be revisited, a posterior over deterministic policies exhibits policy-level commitment, whereas an entropy-regularized stochastic policy resamples actions on each visit. Throughout this subsection the entropy coefficient is set to one.

**One-step stochastic bandit.** Consider a one-step decision problem with initial state $s_3$, actions $a_1, a_2$, and terminal states $s_1, s_2$. Action $a_i$ moves deterministically to $s_i$, where a Bernoulli reward is obtained:

$$R_i = \begin{cases} 1, & \text{with probability } r_i, \\ 0, & \text{with probability } 1 - r_i. \end{cases} \tag{18}$$

Thus the expected return of choosing $a_i$ is

$$J(a_i) = \mathbb{E}[R_i] = r_i. \tag{19}$$

Let $\pi_i$ denote the deterministic policy that chooses $a_i$ at $s_3$. Under the policy-posterior construction, the unnormalized density assigned to $\pi_i$ is

$$\tilde{p}(\pi_i) = \exp J(\pi_i) = \exp(r_i). \tag{20}$$

The posterior-induced action probabilities are therefore

$$p^\star(a_1 \mid s_3) = \frac{\exp(r_1)}{\exp(r_1) + \exp(r_2)} \tag{21}$$

and

$$p^\star(a_2 \mid s_3) = \frac{\exp(r_2)}{\exp(r_1) + \exp(r_2)}. \tag{22}$$

Entropy-regularized RL gives the same probabilities in this one-step setting. Writing $p$ for the probability of choosing $a_1$, the entropy-regularized objective is

$$\max_{p \in [0,1]} \left[ pr_1 + (1-p)r_2 - p \log p - (1-p) \log(1-p) \right]. \tag{23}$$

The optimizer is

$$p_{\text{ER}}(a_1 \mid s_3) = \frac{\exp(r_1)}{\exp(r_1) + \exp(r_2)} = p^\star(a_1 \mid s_3), \tag{24}$$

and similarly for $a_2$. Thus, in a one-step bandit, the policy-posterior construction and entropy-regularized RL coincide.

Optimality-variable control-as-inference differs in the presence of stochastic rewards. In that construction, the unnormalized weight of action $a_i$ is obtained by exponentiating the realized reward and then taking expectation:

$$\mathbb{E}\left[\exp(R_i)\right] = r_i \exp(1) + (1 - r_i) \exp(0) = r_i e + (1 - r_i). \tag{25}$$

Hence the corresponding action probability is

$$p_{\text{CAI}}(a_1 \mid s_3) = \frac{r_1 e + (1 - r_1)}{r_1 e + (1 - r_1) + r_2 e + (1 - r_2)}. \tag{26}$$

This is generally different from

$$\frac{\exp(r_1)}{\exp(r_1) + \exp(r_2)}. \tag{27}$$

The one-step bandit therefore separates the policy-posterior and entropy-regularized objectives from optimality-variable control-as-inference, while also showing that the former two need not differ in the simplest setting.

The same calculation applies to any finite one-step bandit: for actions $a_1, \ldots, a_K$ with expected rewards $\mu_i = \mathbb{E}[R_i]$, both the policy-posterior construction and entropy-regularized RL assign

$$p(a_i) = \frac{\exp(\mu_i)}{\sum_{j=1}^K \exp(\mu_j)}. \tag{28}$$

**Stochastic continuation.** We next give an example in which the policy-posterior construction and entropy-regularized RL differ. Let $s_1$ be the initial state and let $s_2$ be terminal. There are two actions at $s_1$. Action $a_1$ gives zero reward and either returns to $s_1$ or terminates:

$$\mathcal{T}(s_1 \mid s_1, a_1) = \rho, \qquad \mathcal{T}(s_2 \mid s_1, a_1) = 1 - \rho, \qquad R(s_1, a_1, s_1) = R(s_1, a_1, s_2) = 0. \tag{29}$$

where $0 \le \rho < 1$. Action $a_2$ terminates immediately and gives reward one:

$$\mathcal{T}(s_2 \mid s_1, a_2) = 1, \qquad R(s_1, a_2, s_2) = 1. \tag{30}$$

There are two deterministic policies at $s_1$:

$$\pi_1(s_1) = a_1, \qquad \pi_2(s_1) = a_2. \tag{31}$$

Under $\pi_1$, every realized trajectory receives total reward zero. The transition probability $\rho$ affects the number of visits to $s_1$, but it does not affect the total reward. Thus

$$J(\pi_1) = 0. \tag{32}$$

Under $\pi_2$, the process terminates immediately with reward one, so

$$J(\pi_2) = 1. \tag{33}$$

Thus

$$\tilde{p}(\pi_1) = \exp(0) = 1, \qquad \tilde{p}(\pi_2) = \exp(1) = e. \tag{34}$$

Therefore the posterior-induced action probabilities are

$$p^\star(a_1 \mid s_1) = \Pr_{\pi \sim p}[\pi(s_1) = a_1] = \frac{1}{1 + e}, \tag{35}$$

and

$$p^\star(a_2 \mid s_1) = \Pr_{\pi \sim p}[\pi(s_1) = a_2] = \frac{e}{1 + e}. \tag{36}$$

These probabilities are independent of $\rho$.

Entropy-regularized RL gives a different stationary policy. Let $V_{\mathrm{ER}}(s_2) = 0$, and write

$$V := V_{\mathrm{ER}}(s_1). \tag{37}$$

The soft Bellman equation is

$$V = \log\left(\exp(\rho V) + \exp(1)\right), \tag{38}$$

because the soft value after taking $a_1$ is returned to with probability $\rho$, whereas $a_2$ terminates with reward one. Let

$$x := \exp(V). \tag{39}$$

Then $x$ satisfies

$$x = x^\rho + e. \tag{40}$$

The entropy-regularized action probabilities are

$$p_{\mathrm{ER}}(a_2 \mid s_1) = \frac{\exp(1)}{\exp(\rho V) + \exp(1)} = \frac{e}{x}, \tag{41}$$

and

$$p_{\mathrm{ER}}(a_1 \mid s_1) = \frac{\exp(\rho V)}{\exp(\rho V) + \exp(1)} = \frac{x^\rho}{x}. \tag{42}$$

For $\rho = 0$, the example reduces to the one-step case and the two policies coincide. For every $\rho > 0$, however, $x^\rho > 1$ and hence

$$x = x^\rho + e > 1 + e. \tag{43}$$

Consequently,

$$p_{\text{ER}}(a_2 \mid s_1) = \frac{e}{x} < \frac{e}{1 + e} = p^{\star}(a_2 \mid s_1). \tag{44}$$

For example, when $\rho = 1/2$, the equation

$$x = x^{1/2} + e \tag{45}$$

has solution

$$x = \left( \frac{1 + \sqrt{1 + 4e}}{2} \right)^2. \tag{46}$$

Thus

$$p_{\text{ER}}(a_2 \mid s_1) = \frac{e}{\left( \frac{1+\sqrt{1+4e}}{2} \right)^2} \approx 0.55, \tag{47}$$

whereas

$$p^{\star}(a_2 \mid s_1) = \frac{e}{1 + e} \approx 0.73. \tag{48}$$

This example isolates the relevant distinction. The policy-posterior construction places a posterior distribution over deterministic policies. A sampled policy that chooses $a_1$ at $s_1$ is therefore committed to choosing $a_1$ whenever $s_1$ is revisited. Entropy-regularized RL instead optimizes a single stochastic policy. If that policy chooses $a_1$ and the process returns to $s_1$, the next action is resampled. The difference is therefore one of policy-level commitment versus action-level resampling.

## 4 Inference

We perform posterior inference over **deterministic policies**. The formulation also admits inference over stochastic policies, but we focus on deterministic policies because, for the finite-horizon MDP objective considered here, there exists an optimal deterministic policy. In our formulation, stochasticity in control is therefore not introduced by treating the policy itself as intrinsically stochastic; instead, it arises from posterior uncertainty over which deterministic policy is optimal. This keeps uncertainty at the policy level and avoids adding a separate layer of action-level randomness. The stochastic control rule is the posterior-induced policy $p^{\star}(a \mid s)$ defined in Section 3.1; in practice, we use the variational approximation $q(a \mid s)$ and sample actions from $q(\cdot \mid s)$ at execution time.

A natural baseline for posterior inference under the rollout estimator in (16) is structured variational inference (Hoffman et al., 2013), but single-trajectory objectives are often underdispersed and prone to mode collapse (Blei et al., 2017; Yao et al., 2018). The sequential structure of returns makes variational sequential Monte Carlo (VSMC) a natural alternative: its multi-particle objective gives a tighter variational bound and a more robust particle approximation. In our setting, the expected return $J(\pi)$ is available only through the unbiased Monte Carlo estimator (16). This estimator can be treated as exogenous estimator noise on the target density, analogous to pseudo-marginal inference (Andrieu & Roberts, 2009), and incorporated directly into VSMC optimization.

We assume a countable state space and a finite action space to enable revisit bookkeeping and categorical action proposals; these assumptions simplify the inference mechanics and do not affect the probabilistic model. For deterministic policy inference, a policy assigns a single action to each visited state, sampled on first visit:

$$\pi(s) \equiv a \mid s \sim \text{Categorical}(\boldsymbol{p}(s)). \tag{49}$$

In our case studies, $\boldsymbol{p}(s)$ is parameterized by a neural network over a factored representation of $s$.

**SMC sweep.** Two adjustments to the vanilla SMC sweep are required:

1. **Deterministic policy consistency.** For each particle, the action for a state is sampled from the proposal only on the first visit to that state and is reused on all revisits. Equivalently, the particle memoizes $\pi(s)$; on revisits, the proposal assigns probability 1 to the previously sampled action and 0 to all others.

2. **Coupled transition randomness.** To ensure particle weights reflect policy differences rather than independent realizations of environment noise, transition randomness is shared across particles within a sweep. Specifically, if two particles visit the same state $s$ and take the same action $a$ on the same visit count $k$, they are forced to transition to the same successor state $s'$. This can be implemented by lazily sampling and caching $\hat{T}_{s,a}^k \sim \mathcal{T}(\cdot \mid s, a)$ once per queried $(s, a, k)$ and reusing it for all particles in the sweep, so that inference proceeds under a shared random realization $\hat{T}$ of the dynamics.[1]

With these modifications, each sweep proceeds as in standard SMC: particles advance under STEP, update weights, and resample as needed. Full pseudocode appears in the appendix. Proposition 2 specializes Proposition 1 to policy inference.

**Proposition 2** (Policy-VSMC target). *Let $A_s = \pi(s)$ denote the latent action assignment of a deterministic policy at state $s$. Under the posterior $p(\pi)$ induced by Eq. (15), the ideal sequential proposal for a first visit to state $s_t$ is*

$$q^\star(a \mid s_t, \pi_{<t}) = \Pr\left[A_{s_t} = a \mid \pi \text{ agrees with } \pi_{<t}\right], \tag{50}$$

*where $\pi_{<t}$ denotes the action assignments fixed on previous first visits. Marginalizing over the previous assignments gives the posterior-induced stochastic policy*

$$p^\star(a \mid s) = \Pr[\pi(s) = a]. \tag{51}$$

*Thus the learned state-indexed proposal provides an amortized (shared across SMC sweeps and state visits) approximation to the conditional proposal during SMC and to the marginal $p^\star(\cdot \mid s)$ at execution time.*

*Proof.* A deterministic policy is equivalently the collection of latent action assignments $\{A_s : s \in \mathcal{S}\}$. By the first-visit construction above, Policy VSMC samples such an assignment only on the first visit to $s$ and reuses it thereafter. Hence the sequential latent variables are the first-visit policy assignments. Applying Proposition 1 to this sequential representation gives the posterior conditional in Eq. (50). The Markov property of the MDP ensures that, once the current state and the previously fixed policy assignments are given, future rewards depend on the past trajectory only through these quantities. Thus the next proposal variable is the policy assignment $A_{s_t}$, not an action indexed by the full history.

The posterior-induced stochastic policy is the marginal probability that a posterior-sampled deterministic policy assigns action $a$ to state $s$. Marginalizing the conditional proposal over previous first-visit assignments therefore gives

$$\Pr[A_s = a] = \Pr[\pi(s) = a] = p^\star(a \mid s), \tag{52}$$

which proves the claim. $\qquad\square$

At execution time, we use the learned proposal as an amortized approximation to the posterior-induced stochastic policy. If the VSMC approximation were exact, sampling $\pi$ and executing $\pi(s)$ would sample from $p^\star(\cdot \mid s)$. The learned proposal $q_\theta(a \mid s)$ is trained to approximate the corresponding posterior action conditionals, so we execute by sampling from $q_\theta(\cdot \mid s)$ as an approximation to $p^\star(\cdot \mid s)$.

**Optimization objective** The original VSMC formulation assumes reparameterizable proposals and typically omits score-function terms associated with the non-differentiable *resampling* operation due to their high variance. In finite-action MDPs, however, the categorical proposal over actions is not reparameterizable. Consequently, while the resampling-induced score terms may still be dropped, the score-function contribution from *sampling actions from the proposal* must be retained to obtain meaningful gradients. Using a temporally stratified, variance-reduced learning signal (Schulman et al., 2015), we optimize

$$\mathcal{L} = \log \hat{Z} + \sum_{t=1}^{H} \left( \overline{\log \hat{Z}_t} \cdot \sum_{i=1}^{N} \log q_\theta(a_{t,i} \mid s_{t,i}) \right), \tag{53}$$

---

[1]This is related to common random numbers (Mohamed et al., 2020), but is used here to preserve policy-level comparisons across particles rather than only for variance reduction.

where $\log \hat{Z}_t$ denotes the contribution of steps $t, \ldots, H$ to $\log \hat{Z}$, and the overline denotes a stop-gradient operation.

Proposition 3 shows that optimizing the surrogate objective corresponds to stochastic gradient ascent on a well-defined scalar objective, rather than a heuristic update rule.

**Proposition 3** (Score-function gradient with stopped resampling)**.** *Let $\hat{Z}(\hat{T}, \mathbf{a}, \mathbf{r})$ denote the SMC normalizing constant estimator produced by one sweep of the procedure above, where $\hat{T}$ is the shared random realization of the transition dynamics induced by the coupled-transition rule, $\mathbf{a} = \{a_{t,i}\}_{t=1,i=1}^{H,N}$ are the actions sampled from the proposal $q_\theta$, and $\mathbf{r}$ denotes the resampling randomness. Define*

$$\mathcal{J}(\theta) = \mathbb{E}_{\hat{T}, \, \mathbf{a} \sim q_\theta, \, \mathbf{r}} \left[ \log \hat{Z}(\hat{T}, \mathbf{a}, \mathbf{r}) \right]. \tag{54}$$

*Then, treating resampling decisions as stopped-gradient randomness, the gradient of the surrogate objective in Eq. (53) is an unbiased estimator of the gradient of this stopped-resampling objective with respect to the proposal parameters $\theta$.*

*Proof.* The SMC sweep contains three sources of randomness: the shared transition realization $\hat{T}$, the proposal-sampled actions $\mathbf{a}$, and the resampling decisions $\mathbf{r}$. As in standard VSMC, we do not differentiate through the discrete resampling operation and omit the corresponding resampling score-function terms. The claim is therefore about the objective obtained after this standard VSMC approximation, with gradients taken through the action proposal.

Conditioned on $\hat{T}$ and $\mathbf{r}$, the remaining $\theta$-dependent randomness in $\log \hat{Z}(\hat{T}, \mathbf{a}, \mathbf{r})$ comes from sampling actions from $q_\theta$. The realized value of $\log \hat{Z}$ is then a deterministic function of the sampled actions, the shared transition realization, and the resampling decisions. Applying the score-function identity to the proposal-sampled actions gives

$$\nabla_\theta \mathbb{E}_{\mathbf{a}} \left[ \log \hat{Z}(\hat{T}, \mathbf{a}, \mathbf{r}) \right] = \mathbb{E}_{\mathbf{a}} \left[ \log \hat{Z}(\hat{T}, \mathbf{a}, \mathbf{r}) \sum_{t,i} \nabla_\theta \log q_\theta(a_{t,i} \mid s_{t,i}) \right], \tag{55}$$

where the expectation is conditional on $\hat{T}$ and $\mathbf{r}$. Introducing a stop-gradient baseline preserves unbiasedness of the score-function estimator. Taking expectation over $\hat{T}$ and $\mathbf{r}$ completes the result.

The temporally stratified signal $\log \hat{Z}_t$ in Eq. (53) is a variance-reduction, or Rao–Blackwellization, of the same score-function estimator: each sampled action is paired only with the future contribution to $\log \hat{Z}$ that can depend on it. This leaves the expectation unchanged. $\square$

## 5 Related Work

This work relates to (i) probabilistic formulations of planning and control, and (ii) entropy-regularized reinforcement learning. Our key distinction is that inference is carried out over *policies* themselves, rather than over trajectory-level auxiliary variables or within a directly optimized parametric stochastic policy.

### 5.1 Control and Planning as Inference

Casting control and planning as probabilistic inference has a long history, including planning-as-inference in graphical models (Attias, 2003) and subsequent formulations that encode optimality via auxiliary variables or likelihood terms that bias trajectories toward high return (e.g., Botvinick & Toussaint (2012); Toussaint & Storkey (2006)). Active inference likewise casts action selection as approximate Bayesian inference under a generative model with preferences over outcomes. More recently, control-as-inference derivations have shown that entropy-regularized RL objectives arise from variational inference constructions (e.g., Levine (2018)), typically by introducing fictitious observations or optimality variables.

We adopt this perspective but make a different modeling choice: the *policy itself* is the latent random variable, and its expected return defines an unnormalized log density. This yields a posterior over policies directly, without introducing additional observation channels or trajectory-level optimality variables.

## 5.2 Entropy-Regularized Reinforcement Learning

Entropy-regularized RL and stochastic policy optimization methods, including policy gradients (Sutton et al., 1999), soft Q-learning (Haarnoja et al., 2017), and Soft Actor-Critic (SAC) (Haarnoja et al., 2018; Christodoulou, 2019), optimize objectives of the form $\mathbb{E}[R(\pi)] + \alpha\mathcal{H}(\pi)$. Connections and equivalences among these approaches have been studied extensively (e.g., Schulman et al. (2017)), and from a control-as-inference viewpoint entropy can be interpreted as arising from a variational bound.

While related, our approach differs in two respects. First, stochasticity reflects posterior uncertainty over deterministic policies, rather than entropy within a single learned stochastic policy. Second, this inference formulation makes **variational sequential Monte Carlo (VSMC)** (Naesseth et al., 2018) a natural approximation method, in contrast to the single-trajectory variational objectives that underlie many entropy-regularized RL algorithms.

## 5.3 Risk-sensitive and soft-robust MDPs

Risk-sensitive and soft-robust MDPs provide a related but distinct comparison point. A common risk-sensitive objective is the entropic risk measure of cumulative cost,

$$\log \mathbb{E}_{T,\pi}\left[\exp\left(\alpha\sum_{t=1}^{H} c_t\right)\right],\tag{56}$$

which has recently been connected to soft-robust MDPs and studied algorithmically (Zhang et al., 2024; Marthe et al., 2025). With rewards rather than costs, the analogous exponentiated-return objective coincides with the optimality-variable control-as-inference objective (Levine, 2018):

$$\log \mathbb{E}_{T,\pi}\left[\exp\left(\sum_{t=1}^{H} R_t\right)\right].\tag{57}$$

The distinction from our policy-posterior construction is the order in which environment stochasticity and policy uncertainty are handled. Entropic-risk and optimality-variable control-as-inference objectives exponentiate realized trajectory returns and then marginalize over the stochastic quantities, *effectively treating stochastic transitions as latent model variables*. In contrast, our construction first averages environment stochasticity when evaluating a fixed deterministic policy,

$$J(\pi) = \mathbb{E}_T\left[\sum_{t=1}^{H} R_t\right],\tag{58}$$

and then assigns policy density $\tilde{p}(\pi) = \exp\{J(\pi)\}$. The posterior stochastic policy is obtained by marginalizing over deterministic policies. As the worked tabular example shows, these constructions can coincide in simple deterministic settings but generally differ in stochastic MDPs. Thus, entropic-risk methods intentionally modify the control objective to control sensitivity to high-cost or high-reward outcomes. Our objective is different: we retain expected-return evaluation for each deterministic policy and use inference to represent uncertainty over which deterministic policy is selected.

Risk-sensitive RL is therefore not a direct algorithmic baseline for the experiments in this paper. Its purpose is to change the criterion used to evaluate a policy under environment uncertainty, for example by making the agent optimistic or pessimistic with respect to stochastic transitions and rewards. This work instead assumes that the environment model is fixed and uses a posterior distribution over policies to represent uncertainty over agent preferences.

# 6 Experiments

We evaluate the proposed policy inference framework across a range of domains designed to expose different structural aspects of decision making under uncertainty, and compare it to entropy-regularized policy

optimization. Throughout the experiments we contrast inference over distributions of deterministic policies (VSMC) with direct optimization of entropy-regularized stochastic policies (SAC). The experiments are designed to examine (i) qualitative structure of induced behavior in a diagnostic domain, and (ii) differences between deterministic-policy inference and entropy-regularized optimization across increasingly stochastic and complex planning problems. The purpose of this comparison is not to establish uniform superiority in expected reward, since the two methods optimize different objectives, but to examine how these objectives induce different solution structure, uncertainty representations, and action-selection behavior across domains. The experiments are intended to probe the consequences of this modeling choice—inference over policies—rather than to position VSMC as a new state-of-the-art optimizer for entropy-regularized control benchmarks.

We begin by exploring the proposed policy inference framework on a grid world domain. The ease of static visualization and apparent simplicity of grid worlds facilitates qualitative inspection of the induced stochastic policy. We then use the grid worlds and three standard discrete benchmarks from the literature—Blackjack, Triangle Tireworld, and Academic Advising—to compare policy VSMC to discrete Soft Actor-Critic (SAC)[2] (Haarnoja et al., 2018; Christodoulou, 2019), highlighting differences in the resulting policies and their suitability to particular MDP types.

Throughout the experiments, we run VSMC with 10 particles for 50,000 iterations (SMC sweeps), adjusting the initial learning rate between $10^{-5}$ and $3 \cdot 10^{-4}$ for each domain, with cosine decay to 0.1 of the initial rate. These settings are reported to support reproducibility rather than to suggest an "optimal" tuning: in our experiments VSMC is stable across a relatively broad range of learning rates and schedules, the iteration budget is chosen to be large enough to ensure convergence across all domains, and we intentionally keep the number of particles small. This is consistent both with the original VSMC results of Naesseth et al. (2018), which show strong performance with small particle counts, and with the broader particle-variational-inference analysis of Rainforth et al. (2018), which shows that increasing the number of particles can tighten the bound while degrading the signal-to-noise ratio of inference-network gradients. We adapt SAC from CleanRL (Huang et al., 2022) for discrete actions by using MLP critics with two hidden layers, and train for 1,000,000 time steps. In both VSMC and SAC, all networks use two hidden layers of width 64. Each algorithm–domain pair is evaluated over 25 independent training runs, and policies are evaluated using 10,000 trajectories per run. The entropy weight for SAC is kept at $\alpha = 1$ except where varied explicitly. We fix $\alpha$ rather than tuning a domain-specific entropy-weight schedule because the experiments are intended to compare modeling choices—posterior inference over a distribution of deterministic policies versus direct optimization of a stochastic policy—rather than optimize SAC hyperparameters for each benchmark.

### 6.1 Grid Worlds

Grid worlds provide a controlled setting in which policy distributions can be visualized directly, allowing qualitative inspection of multimodality, uncertainty, and variability across runs. They therefore serve as a diagnostic domain for understanding the behavior of the inference procedure itself.

In the grid world domain we use in this study, the environment is a rectangular grid. Four actions—Right, Up, Down, and Left—advance the agent to the corresponding adjacent cell. An action that would take the agent out of the grid has no effect. The environment is slippery: upon any action the agent moves, with probability $p_{succ} = 0.8$, in the direction of the action, and otherwise in an adjacent direction. The reward collected by the agent upon each action is determined by the color of the cell to which the agent moves: grey (pavement) — 0, red (gravel) — -1, yellow (goal) — +5, green (swamp) — -5. Each step incurs a cost of -0.1. Yellow and green cells are absorbing: once the agent reaches a yellow or a green cell, no further reward is collected and no action moves the agent out of the cell.

Inferred policies are represented by *policy and occupancy maps* in the figures below (Figure 1–3). A black border denotes the initial cell. The darkness of the middle part of a cell represents the occupancy: the

---

[2]SAC optimizes an entropy-regularized expected-return objective, whereas policy VSMC maximizes an SMC log-evidence bound $\mathbb{E}[\log \hat{Z}]$, where $\log \hat{Z}$ aggregates particle weights via a log-mean-exp across the sweep. The two objectives are not the same, so differences in performance are informative primarily as differences in induced behavior rather than as a pure leaderboard comparison.

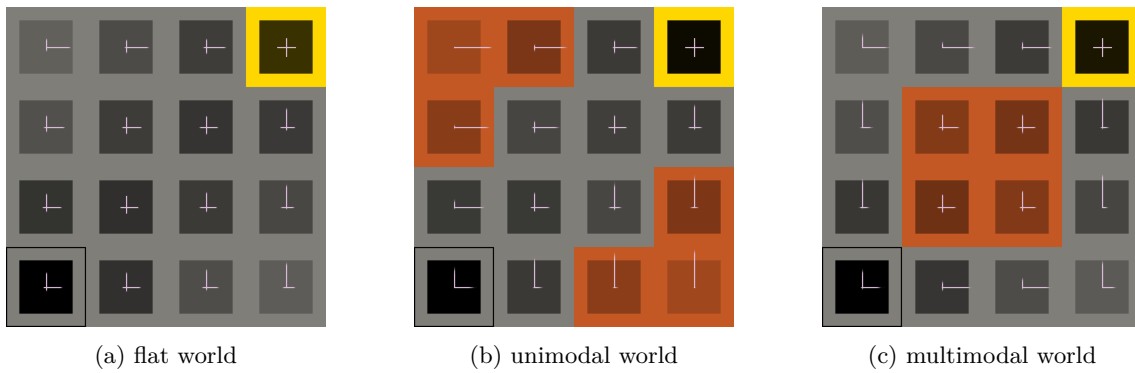

(a) flat world      (b) unimodal world      (c) multimodal world

Figure 1: Grid worlds: policies and occupancies

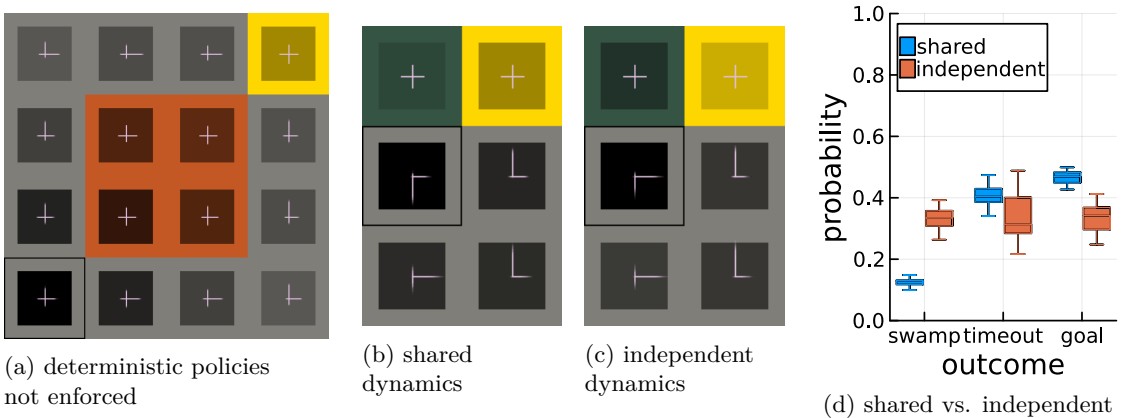

(a) deterministic policies not enforced    (b) shared dynamics    (c) independent dynamics    (d) shared vs. independent

Figure 2: Grid worlds: ablation studies

darker a cell, the more trajectories passed through the cell. The white cross with unevenly sized beams in the center of each cell represents the policy distribution in that cell, with the length of each beam denoting the probability of the corresponding direction. The maps are averaged over 25 runs: blurrier crosses mean more variation in the inferred policies across runs.

We begin by applying policy VSMC to three $4 \times 4$ grid worlds (Figure 1), using a rollout horizon of 20 steps. In the flat world (Figure 1a), trajectories cover the grid evenly. In the "unimodal" world (Figure 1b) the expensive to travel red regions in the complementary corners of the grid push trajectories closer to the diagonal connecting the starting and the goal cells. In the "multimodal" world (Figure 1c) the red region is in the center of the grid, pushing the trajectories to pass along the edges of the grid. Because a policy *distribution* is inferred (rather than just a single policy with the highest expected return), multiple actions in each cell have non-zero probabilities.

For policy inference, the SMC sweep of VSMC was modified to a) enforce deterministic policies b) share the same environment dynamics among all particles. Figure 2a shows the effect of dropping the enforcement of deterministic policies: a higher-entropy policy distribution. Figures 2b–2d explore the effect of sharing environment dynamics among all particles on the inferred policies. A small slippery instance with two absorbing cells, a swamp with reward of -5 at (1, 3) and a goal with reward of 5 at (2, 3), and $p_{succ} = 0.5$ is used for the comparison. The starting position is at (1, 2) and the rollout horizon is 10 steps. To avoid slipping into the swamp, the agent should move *down*, to (1, 1), and this is what the policy with shared dynamics mostly suggests. With independent dynamics the agent frequently moves *right*, along the shortest path to the goal.

Finally, we compare VSMC to SAC. Figure 3 shows the policies inferred by VSMC (Figure 3a) and SAC (Figure 3b) and compares their trajectory return distributions (Figure 3c). VSMC and SAC trajectory

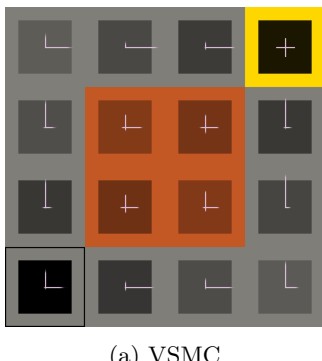

(a) VSMC

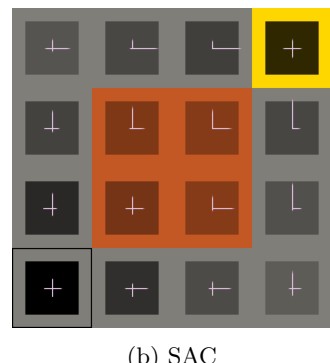

(b) SAC

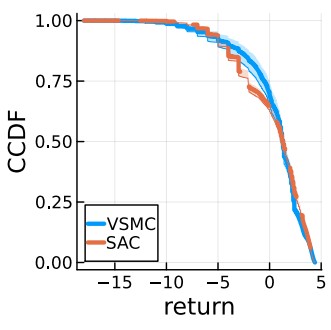

(c) trajectory return distributions

Figure 3: Grid worlds: VSMC vs. SAC

Table 1: Blackjack: policy statistics

| policy | expected return | probability | | |
|---|---|---|---|---|
| | | loss | draw | win |
| VSMC | $-0.19 \pm 0.04$ | $0.57 \pm 0.02$ | $0.05 \pm 0.01$ | $0.38 \pm 0.02$ |
| VSMC, $10 \cdot r$ | $-0.14 \pm 0.03$ | $0.54 \pm 0.01$ | $0.05 \pm 0.01$ | $0.41 \pm 0.01$ |
| VSMC, $100 \cdot r$ | $-0.12 \pm 0.04$ | $0.53 \pm 0.02$ | $0.06 \pm 0.01$ | $0.41 \pm 0.02$ |
| SAC | $-0.32 \pm 0.01$ | $0.63 \pm 0.01$ | $0.06 \pm 0.01$ | $0.31 \pm 0.01$ |
| SAC, $\alpha = 0.1$ | $-0.15 \pm 0.01$ | $0.54 \pm 0.02$ | $0.07 \pm 0.01$ | $0.39 \pm 0.02$ |
| SAC, $\alpha = 0.01$ | $-0.08 \pm 0.03$ | $0.50 \pm 0.05$ | $0.08 \pm 0.02$ | $0.42 \pm 0.03$ |
| optimal | $-0.06$ | $0.49$ | $0.09$ | $0.42$ |

return distributions are close but different, and the policies differ in particular along the grid edges — SAC, optimizing an entropy-regularized stochastic policy, uses actions directed toward the grid boundaries to increase the entropy. VSMC penalizes such actions strongly because a deterministic policy directing the agent into a grid boundary can escape the current cell only due to environment stochasticity.

## 6.2 Blackjack

Blackjack is a card game where the goal is to beat the dealer by obtaining cards that sum to closer to 21 (without going over 21) than the dealer's cards. Blackjack provides a stochastic control problem with a compact state space and a known optimal policy, making it possible to compare inferred policies against a ground-truth solution while examining the effect of entropy regularization in a domain where randomness arises primarily from the environment rather than exploration. The game starts with the dealer having one face up and one face down card, while the player has two face up cards. All cards are drawn from an infinite deck (i.e. with replacement). The player has the sum of cards held. The player can request additional cards (hit) until they decide to stop (stick) or exceed 21 (bust). After the player sticks, the dealer reveals their facedown card, and draws cards until their sum is 17 or greater. If the dealer goes bust, the player wins. If neither the player nor the dealer busts, the outcome (win, lose, draw) is decided by whose sum is closer to 21.

The variant described in Sutton & Barto (2018, Example 5.1), as implemented in Gymnasium (Towers et al., 2024), is used in this study. VSMC and SAC are compared to each other and to the optimal policy, as a baseline, in Table 1 and Figure 4. The optimal (return-maximizing) player's policy was computed by value iteration and the policy's statistics were estimated by Monte-Carlo evaluation. Neither VSMC nor SAC with the default temperature/entropy weight matches the optimal policy, which is expected because both methods are deliberately regularized. The main observation is not that one method dominates the other in mean return, but that the two objectives induce different operating points: VSMC produces a lower draw probability than SAC at $\alpha = 1$, while matching this behavior in SAC requires substantially weaker entropy regularization ($\alpha = 0.1$ or $\alpha = 0.01$). At $\times 10$ and $\times 100$ higher reward magnitudes, VSMC approaches the

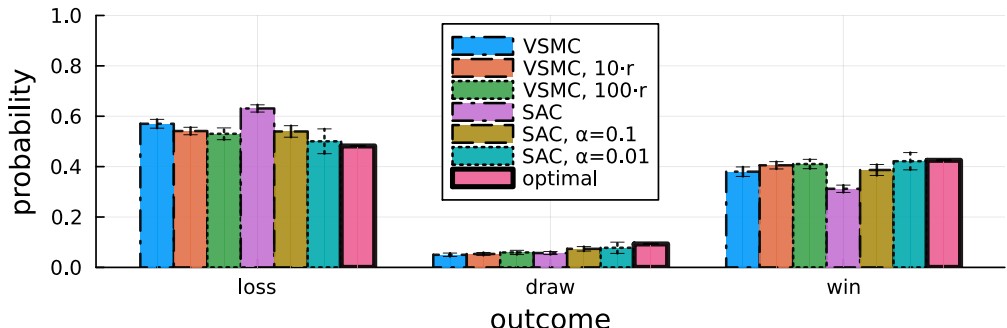

Figure 4: Blackjack: outcome probabilities

Table 2: Triangle Tireworld: policy statistics

| # | policy | expected return | success probability |
|---|--------|-----------------|---------------------|
|   | VSMC | $-0.80 \pm 2.38$ | $0.47 \pm 0.12$ |
| 1 | VSMC $(0.2 \cdot r)$ | $1.25 \pm 0.13$ | $0.58 \pm 0.02$ |
|   | SAC | $1.27 \pm 0.11$ | $0.58 \pm 0.01$ |
|   | VSMC | $-4.39 \pm 1.46$ | $0.30 \pm 0.08$ |
| 4 | VSMC $(0.2 \cdot r)$ | $-1.82 \pm 0.22$ | $0.44 \pm 0.02$ |
|   | SAC | $-1.90 \pm 0.17$ | $0.43 \pm 0.01$ |
|   | VSMC | $-2.91 \pm 1.14$ | $0.39 \pm 0.02$ |
| 7 | VSMC $(0.2 \cdot r)$ | $-1.33 \pm 0.16$ | $0.48 \pm 0.03$ |
|   | SAC | $-1.35 \pm 0.09$ | $0.48 \pm 0.01$ |
|   | VSMC | $-5.79 \pm 0.84$ | $0.22 \pm 0.06$ |
| 10 | VSMC $(0.2 \cdot r)$ | $-4.89 \pm 0.19$ | $0.30 \pm 0.02$ |
|   | SAC | $-5.27 \pm 0.11$ | $0.27 \pm 0.01$ |

optimal win probability, while still having a lower draw probability and a higher loss probability than SAC with corresponding entropy regularization.

These results highlight that, under comparable regularization strength, policy inference and entropy-regularized optimization induce different trade-offs between exploration and outcome variance, even in a domain with a known optimal policy.

### 6.3 Triangle Tireworld

In Triangle Tireworld (Little & Thiébaux, 2007), the agent travels through a triangular structure of locations connected by directed roads. The agent must arrive from the initial location to the goal. With a fixed probability, the agent gets a flat. Some locations carry a spare tire. If the agent loaded a spare tire prior to getting a flat, it changes the tire and continues the travel. Otherwise, the agent is stuck. The agent gets the reward of 10 for reaching the goal, -10 for getting stuck, and -0.1 for every step. Triangle Tireworld introduces irreversible stochastic events and explicit risk–reward trade-offs. Successful policies must plan for low-probability failures whose consequences cannot be undone. The domain and the instances are based on the version from the 2014 International planning competition Vallati et al. (2015).

With the original rewards, Triangle Tireworld induces a large return gap between "fast but risky" and "safe but slow" behaviors. Under our Bayesian formulation this makes the posterior highly peaked, yielding low mean return and high variance. Scaling rewards down by a factor of five reduces this separation, producing a less concentrated posterior; under this setting VSMC exhibits performance comparable to SAC. Table 2 and Figure 5 report results for instances 1, 4, 7, and 10, which are representative instances spanning the range of difficulty in this domain; the omitted instances follow the same qualitative trend.

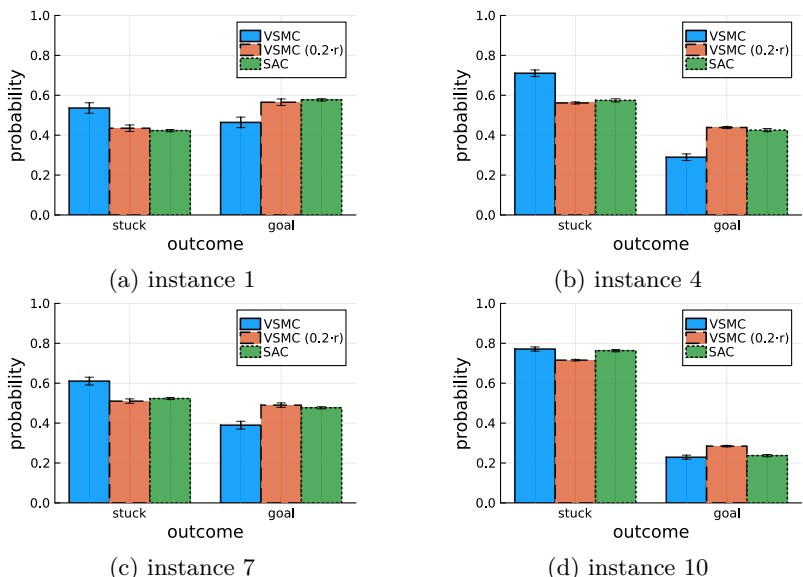

Figure 5: Triangle Tireworld: outcome probabilities

Table 3: Academic Advising: policy statistics

| # | policy | expected return | 0.05 | | 0.95 | |
|---|---|---|---|---|---|---|
| | | | quantile | tail mean | quantile | tail mean |
| 1 | VSMC | $-65.3 \pm 1.3$ | $-141.1 \pm 3.6$ | $-175.7 \pm 5.0$ | $-20.4 \pm 0.5$ | $-18.4 \pm 0.4$ |
| | SAC | $-48.6 \pm 0.8$ | $-84.0 \pm 2.8$ | $-97.1 \pm 2.6$ | $-24.8 \pm 1.2$ | $-21.9 \pm 0.9$ |
| 2 | VSMC | $-98.5 \pm 1.8$ | $-184.2 \pm 5.2$ | $-222.1 \pm 7.1$ | $-45.0 \pm 1.2$ | $-39.5 \pm 1.0$ |
| | SAC | $-106.7 \pm 3.6$ | $-184.6 \pm 8.2$ | $-216.1 \pm 11.1$ | $-52.8 \pm 1.4$ | $-45.7 \pm 1.2$ |
| 3 | VSMC | $-86.7 \pm 1.1$ | $-174.1 \pm 3.0$ | $-207.5 \pm 3.4$ | $-32.6 \pm 0.2$ | $-28.1 \pm 0.5$ |
| | SAC | $-84.4 \pm 2.2$ | $-147.7 \pm 4.5$ | $-174.6 \pm 5.3$ | $-39.0 \pm 1.6$ | $-34.2 \pm 0.7$ |

Triangle Tireworld highlights a limitation of Bayesian policy inference: unlike classical MDP planning, which is invariant to affine reward scaling, the posterior depends on return magnitudes, so the method works best when reward scale meaningfully encodes the strength of preferences/regrets rather than merely ranking policies.

## 6.4 Academic Advising

Academic Advising models a student choosing which courses to take over a sequence of semesters in order to complete a curriculum. Academic Advising represents a large combinatorial planning problem with long horizons and delayed rewards. The branching action space and stochastic course outcomes create highly multimodal trajectory returns, providing a test of scalability and behavior under complex long-term dependencies. At each semester the action is to enroll in up to a maximum course load of currently eligible courses. Course outcomes are stochastic: each enrolled course is passed with a given probability (otherwise it remains incomplete and can be retaken later), and passed courses persist until curriculum completion, at which point the process enters an absorbing goal state. The reward is specified as step costs: taking a course incurs -1, retaking a previously attempted course incurs -2, and if the program is not yet complete the agent also incurs an additional -5 at every step, encouraging completion before rollout truncation. In the IPC 2014-derived instances we use, the curriculum contains 10–30 courses; easier (odd-numbered) instances restrict the course load to at most 1 course per step, while harder (even-numbered) instances allow up to 2 courses per step.

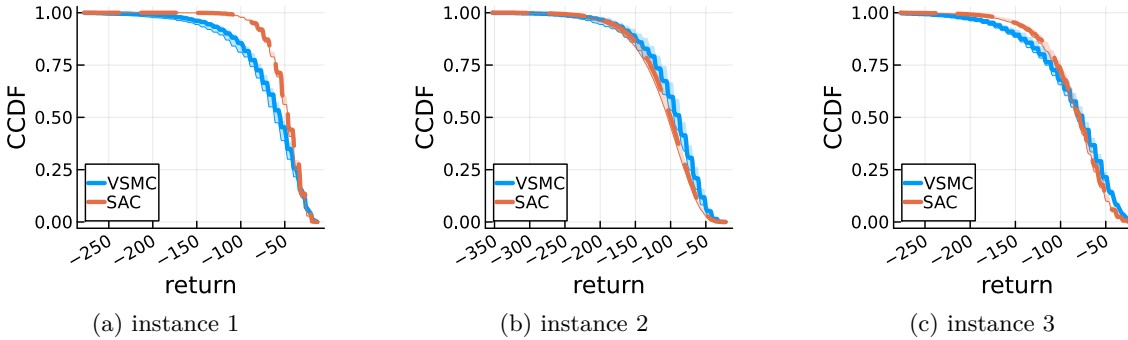

(a) instance 1      (b) instance 2      (c) instance 3

Figure 6: Academic Advising: trajectory return distributions

Without a non-trivial baseline policy or a domain-specific heuristic, SAC and VSMC reliably find policies with a non-negligible probability of completing the program for instances 1–3. For harder instances, either the variation between runs is very high, or the policy / policy distribution concentrates around a random walk that minimizes the per-step cost but does not lead to program completion. Table 3 summarizes policy statistics and Figure 6 shows trajectory return distributions for instances 1–3. The comparison is most informative at the level of return distributions rather than mean return alone: VSMC and SAC often achieve comparable average performance, but VSMC induces heavier-tailed trajectory return distributions, as manifested by the 0.05 and 0.95 quantiles and their conditional tail means. This is consistent with the broader claim of the paper that the main difference lies in how the two objectives represent and act under trajectory uncertainty, not in uniform reward dominance.

The Academic Advising results demonstrate that the differences between the two approaches persist in larger combinatorial settings, where long horizons and stochastic outcomes amplify differences in how trajectory uncertainty is represented.

## 7  Discussion

We view episodic MDP planning as posterior inference over policies, with expected return defining an unnormalized posterior density (Eq. (15)). The resulting posterior concentrates on return-maximizing policies, while its dispersion captures uncertainty about optimal behavior.

**Sources of uncertainty.** With a posterior over deterministic policies, three sources of uncertainty are disentangled:

(i) **aleatoric** transition randomness, sampled forward and appearing as noise in the Monte Carlo estimate of policy log-probability (Eq. (16));

(ii) **epistemic** uncertainty over optimal behavior, represented by posterior dispersion; and

(iii) **execution-time stochasticity**, obtained by marginalizing over deterministic policies. In exact form, posterior-predictive control is therefore a structured form of Thompson sampling: actions randomize only to the extent that multiple deterministic behaviors remain plausible.

**Inference mechanics.** Once planning is cast as inference with an intractable target density, the sequential structure of returns makes SMC a natural fit, and VSMC provides a principled variational objective. Policy inference, however, requires two adaptations: enforcing policy consistency under revisitation (by sampling each state's action only on first visit) and coupling transition randomness across particles within a sweep so that weights reflect policy differences rather than independent simulator noise.

**Relation to control-as-inference and entropy-regularized RL.** Compared to trajectory-centric control-as-inference formulations, our latent variable is the policy itself. Compared to entropy-regularized

RL, stochasticity reflects posterior uncertainty over deterministic behaviors rather than an explicit entropy preference.

**Empirical takeaways.** Across domains, the comparison to SAC should be read primarily as a comparison of objectives and the behaviors they induce, rather than as a claim that policy VSMC is a uniformly stronger optimizer of expected reward. The worked examples show that policy-posterior inference and entropy-regularized optimization can coincide in simple one-step settings, but need not coincide when stochastic continuation or state revisitation makes policy-level commitment differ from action-level resampling. The empirical results are consistent with this distinction, while also reflecting the effects of approximation, optimization, and reward scale. In grid worlds, the variationally approximated induced policy avoids boundary-directed actions that can increase entropy under SAC while reducing goal reachability. This illustrates the basic mechanism behind the empirical differences: entropy-regularized RL randomizes at the action level, whereas policy inference places posterior mass on complete deterministic policies whose repeated statewise choices must themselves be plausible under the return model. In Blackjack, matching VSMC-like behavior requires substantially weaker entropy regularization in SAC, suggesting that the default entropy-regularized objective occupies a different operating point. In Triangle Tireworld, the sensitivity of VSMC to reward scale reveals a substantive property of the Bayesian formulation: return magnitudes control posterior concentration. In Academic Advising, the main contrast is distributional: both methods can achieve similar mean performance on some instances, but they differ in tail behavior and variability. Taken together, these results support the narrower conclusion that policy inference and entropy-regularized optimization induce qualitatively different behaviors even when aggregate returns are similar, rather than a uniform dominance claim for either method.

**Scalability considerations.** The method inherits the usual scaling of particle inference: runtime grows with the rollout horizon and the number of particles. The additional bookkeeping for policy and transition consistency adds memory proportional to the number of distinct state–action–visit-count queries encountered during a sweep. This overhead is modest in the discrete domains considered here, where revisitation occurs, but may become relevant in very large state spaces or domains with little revisitation. This cost is distinct from resampling itself: prior VSMC results (Naesseth et al., 2018) show that resampling can make better use of small particle budgets than IWAE-style variational importance sampling. Overall, scalability is governed primarily by the standard tradeoffs among particle count, rollout length, and effective state-space size.

**Scope and extensions.** We focus on discrete state spaces to make revisit bookkeeping and shared transition caching explicit; the policy-inference semantics do not depend on discreteness. In continuous domains, determinism can be enforced via the policy representation or a hashable state abstraction, and shared stochasticity can be implemented via common random numbers or keyed random streams. Finally, strict memoization can be relaxed via *stochastic memoization*, allowing occasional resampling on revisits to reduce the brittleness of committing to an early no-op action.

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

## Algorithmic Details

---

**Algorithm 1** VSMC sweep for deterministic-policy inference. The procedure returns the surrogate objective in Eq. (53). Resampling is shown in the canonical SMC form; the implementation resamples adaptively (ESS threshold of 0.5).

---

1: **procedure** POLICYVSMC($s_1$, STEP, REWARD, $H$, $N$, $q_\theta$)
2:     $M_T \leftarrow \varnothing$
3:     **for** $i \leftarrow 1$ to $N$ **do**
4:         $s^{(i)} \leftarrow s_1$
5:         $M_A^{(i)} \leftarrow \varnothing$                                         ▷ action memoization: $s \mapsto a$
6:         $M_C^{(i)} \leftarrow \varnothing$                                           ▷ counts: $(s, a) \mapsto k$
7:     **end for**
8:     $\ell_{1:H} \leftarrow 0$                                             ▷ per-step log mean weight increments
9:     $g_{1:H} \leftarrow 0$                                     ▷ per-step proposal log-prob terms (first-visit only)
10:     **for** $t \leftarrow 1$ to $H$ **do**
11:         **for** $i \leftarrow 1$ to $N$ **do**
12:             $s \leftarrow s^{(i)}$
13:             **if** $s \in \text{dom } M_A^{(i)}$ **then**                      ▷ revisit: reuse memoized action
14:                 $a \leftarrow M_A^{(i)}(s)$
15:                 $\log p_a \leftarrow 0, \ \log q_a \leftarrow 0$
16:             **else**                                         ▷ first visit: sample and memoize
17:                 $a \sim q_\theta(\cdot \mid s), \ M_A^{(i)}(s) \leftarrow a$
18:                 $\log p_a \leftarrow -\log |A|, \ \log q_a \leftarrow \log q_\theta(a \mid s)$
19:                 $g_t \leftarrow g_t + \log q_a$
20:             **end if**
21:             $k \leftarrow M_C^{(i)}(s, a) + 1$                    ▷ default $M_C^{(i)}(s, a) = 0$ if absent
22:             $M_C^{(i)}(s, a) \leftarrow k$
23:             **if** $(s, a, k) \in \text{dom } M_T$ **then**                ▷ coupled transition randomness
24:                 $s' \leftarrow M_T(s, a, k)$
25:             **else**
26:                 $s' \leftarrow \text{STEP}(s, a)$
27:                 $M_T(s, a, k) \leftarrow s'$
28:             **end if**
29:             $w^{(i)} \leftarrow \text{REWARD}(s, a, s') + \log p_a - \log q_a$
30:             $s^{(i)} \leftarrow s'$
31:         **end for**
32:         $\ell_t \leftarrow \log \sum_{i=1}^{N} \exp(w^{(i)}) - \log N$
33:         $(s^{(1:N)}, M_A^{(1:N)}, M_C^{(1:N)}) \leftarrow \text{RESAMPLE}\big((s^{(1:N)}, M_A^{(1:N)}, M_C^{(1:N)}), w^{(1:N)}\big)$
34:     **end for**
35:     $\log \hat{Z} \leftarrow \sum_{t=1}^{H} \ell_t$
36:     $\log \hat{Z}_{H+1} \leftarrow 0$
37:     **for** $t \leftarrow H$ down to 1 **do**
38:         $\log \hat{Z}_t \leftarrow \ell_t + \log \hat{Z}_{t+1}$                           ▷ suffix sums for Eq. (53)
39:     **end for**
40:     **return** $\log \hat{Z} + \sum_{t=1}^{H} \left( \overline{\log \hat{Z}_t} \cdot g_t \right)$
41: **end procedure**

---

