# OpenReview forum: "MDP Planning as Policy Inference"
_TMLR — Under review for TMLR_

### Review · Reviewer_QZSq · 2026-05-28

**Summary Of Contributions:**

The paper proposes a Bayesian formulation of episodic MDP planning in which the policy itself is the latent variable. Concretely, the authors define an unnormalized policy density
$$
\log \tilde p(\pi) = \mathbb{E}\left[\sum_{t=1}^{H} R(s_t, a_t, s_{t+1})\right] = J(\pi),
$$
so that $\tilde p(\pi) \propto \exp(J(\pi))$, i.e., a Boltzmann distribution over policies whose energy is the standard expected return from the initial state. By marginalizing the posterior over deterministic policies, they obtain an induced stochastic action-selection rule
$ p^\star(a \mid s) = \Pr_{\pi \sim p}\bigl[\pi(s) = a\bigr], $
which is called *the optimal stochastic policy under preference uncertainty*. This is contrasted with trajectory-centric control-as-inference (which, under the standard variational treatment, yields entropy-regularized RL; Levine 2018).

For inference they adapt variational SMC (Naesseth et al. 2018) to discrete MDPs, with two modifications: (i) deterministic-policy consistency under state revisitation (per-particle action memoization), and (ii) coupled transition randomness across particles within a sweep. Theorem 1 shows the surrogate gradient is an unbiased estimator of $\nabla_\theta \mathbb{E}[\log\hat Z]$. Empirical evaluation: grid worlds (with ablations), Blackjack, Triangle Tireworld, and Academic Advising; compared against discrete SAC.

**Strengths.** The conceptual move to Boltzmann distribution over policies rather than over trajectories is clean and genuinely distinct from prior planning-as-inference work. The two SMC modifications are natural for the deterministic-policy choice. The paper is honest about scope and frames the SAC comparison as informative about induced behavior, not a leaderboard. Grid-world ablations cleanly isolate each modification.

**Main weaknesses.** Two central gaps:

- The new object $p^\star(a \mid s)$ is defined but never computed, illustrated, or analytically characterized, so a reader finishes the paper with no concrete picture of what this policy looks like.
- No consistency claim is made for the VSMC scheme. Theorem 1 concerns only the gradient estimator; the paper does not state (even by reference to existing SMC literature) what target the algorithm provably reaches as $N \to \infty$, or whether $q^*$ recovers the stated posterior.

Additionally, the $\log\hat Z$ structure of the VSMC objective is structurally very close to objectives studied in the risk-sensitive / soft-robust MDP literature, which the paper does not currently discuss (see more details below).

**Audience:**

Yes

**Audience Explanation:**

The reframing of planning-as-inference at the policy level (rather than at the trajectory level) is a genuinely new conceptual move within the control-as-inference literature, and the adaptation of VSMC with revisit memoization and coupled transitions is a clean piece of algorithmic work. Researchers in probabilistic planning, particle-based variational inference, and risk-sensitive RL (see below) should find the paper of interest. The conceptual contribution is appropriate for TMLR's scope; the experiments are diagnostic rather than competitive, which is fine here.

**Broader Impact Concerns:**

None. Methodological paper with small-scale benchmarks; no Broader Impact Statement needed.

**Claims And Evidence:**

No

**Claims Explanation:**

The conceptual claim that planning can be cast as Bayesian inference over policies with target $\propto \exp(J(\pi))$ is well-supported. But the two operational claims that follow are not:

(a) **$p^\star$ is a meaningful new object.** The definition is given (Eq. 5) and then immediately set aside for approximate inference. No tabular example, no closed form in any special case, no asymptotic statement about how $p^\star$ behaves as the reward scale changes, no formal contrast with the entropy-regularized softmax. For me as a reader, I can verify that $p^\star$ behaves as the paper suggests only by looking on the experiments, which raises the second issue.

(b) **The algorithm performs approximate inference under that model.** Theorem 1 establishes unbiasedness of the gradient estimator with respect to $\mathbb{E}[\log\hat Z]$. However, it is a statement about stochastic gradient ascent, not about variational inference quality; the paper does not state (i) what the $N\to\infty$ limit of the particle approximation is, or (ii) whether the optimal proposal $q^*$ recovers the stated posterior $\propto \exp(J(\pi))$. The required result seems to be already contained in Naesseth et al. (2018), but it'd be nice to make everything self-contained, especially given additional modification to VSMC.

**Requested Changes:**

Three critical changes, and several smaller items.

### 1. Show what $p^\star(a\mid s)$ actually looks like

After reading Section 3.1 I have no concrete picture of the new object. Please add, after Eq. (5), a worked example on a small tabular MDP (e.g., 2 states, 2 actions, possibly with deterministic dynamics in a first pass) where $p^\star$ can be enumerated exactly over deterministic policies and reported. This would:

- Demonstrate that $p^\star$ is a well-defined stochastic policy and show its support.
- Show how it differs from the entropy-regularized softmax over the same MDP (ideally, for any given temperature).
- Give intuition for what feature of the MDP the action probabilities reflect.

In addition, please give at least one analytical handle: e.g., a closed form for $p^{\star}$ in the bandit or deterministic-dynamics case, or an asymptotic statement about concentration on argmax policies as the reward scale grows. The entropy-regularized softmax has a clean closed form (softmax over soft Q-values); $p^{\star}$ deserves an analogous treatment, even if only in special cases.

### 2. Establish what the VSMC scheme converges to

The paper proposes posterior inference over policies but does not establish that the algorithm performs that inference. Theorem 1 is about unbiasedness of the gradient estimator and says nothing about consistency of the variational approximation. Please add, ideally as a proposition (which may follow with little work from existing SMC theory):

- The $N\to\infty$ limit of the particle approximation, with reference to the relevant SMC consistency literature (Del Moral; Chopin; Naesseth et al.).
- A statement of which target $q$ converges to. If, as I suspect from the $\log\hat Z$ structure, $q$ recovers the posterior under a different density than the stated $\exp(J(\pi))$ — namely one involving the exp-and-log structure of the SMC objective (see §6.3) — then state that explicitly and explain what this means for the interpretation of $p^\star$ at execution time.
- A justification for sampling actions from $q(\cdot \mid s)$ at execution time. Section 4 simply asserts this without argument.
- A clear statement here would also significantly clarify what the algorithm contributes relative to the conceptual model.

### 3. Discuss the connection to risk-sensitive / soft-robust MDPs
The risk-sensitive MDP literature optimizes objectives that are structurally very close to the VSMC objective in this paper. Concretely: the canonical risk-sensitive RL objective is the entropic risk measure of the return,

$$ \log\mathbb{E}_{T,\pi}\left[\exp\Bigl(\sum_t R_t\Bigr)\right], $$

i.e., the log of the moment generating function of the trajectory return. The paper's VSMC objective $\mathbb{E}[\log\hat Z]$ is in the $N\to\infty$ limit seems to be equal to $ \mathbb{E}_{T}\left[f(T)\right]$ for

$$
f(T) =  \log\mathbb{E}_{\pi\mid T}\left[\exp\Bigl(\sum_t R_t\Bigr)\right].
$$

These coincide on deterministic-transition MDPs but generically differ: the two objectives place the outer log on different sides of the expectation over the dynamics $T$.  Thus, it would be very natural to compare the discuss this potential connection with risk-sensitivie MDP literature (see, e.g., Zhang et al 2024 and Marthe et al. 2025) and potentially build an additional bridge to this line of literature, which should strengthen the paper a lot. Additionally, it might be useful to have additional insights on how the optimal policy $p^{\star}$ can be characterized for Q1.


#### References

Zhang, R., Hu, Y., & Li, N. (2024, May). Soft robust mdps and risk-sensitive mdps: Equivalence, policy gradient, and sample complexity. In International Conference on Learning Representations (Vol. 2024, pp. 14655-14669).

Marthe, A., Bounan, S., Garivier, A., & Vernade, C. (2025). Efficient risk-sensitive planning via entropic risk measures. arXiv preprint arXiv:2502.20423.


### 4. Minor concerns


#### Framing of prior work
The paper presents itself as distinct from two alternatives, trajectory-level control-as-inference and entropy-regularized RL, but these are equivalent presentations of the same framework (Levine 2018, which is cited). The contribution would land more cleanly framed as a single contrast: prior work places the Boltzmann over trajectories; this paper places it over policies. Merging Sections 5.1 and 5.2 would help.

#### Notation and minor presentation

* In Eq. (5), $\pi \sim p(\pi)$ read a little awkwardly to me — might $\pi \sim p$ or $\pi \sim p(\cdot)$ be cleaner?
* Eq. (3) uses both $a_t^{(\pi)}$ and $\pi(s_t)$; it might read more smoothly to unify these as $a_t = \pi(s_t)$.
* In Theorem 1, it could help to make explicit which $\theta$ the gradient is taken with respect to, since $\theta$ does not currently appear in Eq. (7).
* There is some risk of confusion between the horizon $H$ (and the transition kernel $T$) and the time index $t$, especially where limits are involved. One option would be to rename the time index from $t$ to $h$, so that it pairs naturally with the horizon $H$ and avoids clashing with other uses of $T$.
* RL-oriented readers might find it helpful to see the SMC objects in Section 2.2 instantiated in the planning setting: e.g., the SMC "step" as one MDP step, the incremental weight as $\exp(R_t)$, and $\hat Z$ as an estimate of $\mathbb{E}_\tau[\exp(\sum R_t)]$ under the proposal. Since $\log\hat Z$ is introduced fairly quickly in Section 2.2 and then becomes central in Section 4, a little more setup there could be very helpful.

#### Underdeveloped framing choices
* The prior currently appears only in Section 4 ("uniform prior over deterministic policies"). For a paper framed around Bayesian inference, it might read more naturally to introduce the prior already in the model section and perhaps to comment briefly on alternative priors (entropy-promoting, sparsity-promoting), which could be an interesting direction.

* The inverse temperature $\beta$ is currently implicit (effectively set to 1). The authors might consider writing $\tilde p(\pi) \propto \exp(\beta J(\pi))$ explicitly: this would surface $\beta$ as a hyperparameter in its own right, would frame the reward-scale sensitivity observed in Triangle Tireworld as a natural consequence of the non-affine-invariance of Boltzmann distributions rather than as a surprise, and would allow a more symmetric comparison with SAC (where $\alpha$ is varied).

#### Terminology
The phrase "preference uncertainty" overloads RLHF/preference-learning terminology, where it refers to uncertainty over a reward model learned from human preferences.

#### Experiments
* The Blackjack figure varies SAC's $\alpha$ but uses a single VSMC; a reward-scale sweep on the VSMC side would make the comparison symmetric.
* For SAC: is the temperature automatically tuned (Haarnoja et al. 2018b) or fixed?
* The Triangle Tireworld and Academic Advising tables would benefit from distributional plots (eCDF/CCDF as in Figure 3c), since the main qualitative claim is about tail behavior.

---

> ### Author Response · Authors · 2026-06-04
>
> We thank the reviewer for the constructive and detailed review. We found the main suggestions helpful and have revised the paper substantially in response.
>
> First, we added worked tabular examples after the model definition. These examples enumerate the induced policy distribution exactly, show the resulting stochastic action rule $p^\star(a\mid s)$, and compare it directly with entropy-regularized control and optimality-variable control-as-inference. We also generalized the initial example to a closed-form Bernoulli bandit setting, which gives an analytic handle on how the induced stochastic policy behaves as a function of the rewards.
>
> Second, we added formal statements clarifying what the VSMC scheme converges to. In particular, the revision now separates two distinct issues: consistency of the particle approximation for a fixed proposal, and convergence of the learned proposal distribution. The latter is not the same as taking the number of particles $N\to\infty$. In practice, VSMC is used as a variational training objective, and prior work (Rainforth et al., 2018) suggests that relatively small numbers of particles can be preferable for learning the proposal. We therefore clarified that the relevant question is not simply the large-$N$ particle limit, but what target is being optimized by the proposal family and how the resulting approximation is used to form the execution-time stochastic policy.
>
> Third, we added a discussion of the relationship to risk-sensitive RL and soft-robust MDPs. We agree that this is closely related to the optimality-variable control-as-inference objective. The revised related-work section now distinguishes policy-level inference, entropy-regularized RL, optimality-variable control-as-inference, and risk-sensitive objectives. We kept entropy-regularized RL and optimality-variable control-as-inference separate because, although closely related historically, they are not identical objects in the comparison made in this paper.
>
> We also revised the notation throughout the paper. We considered all of the notation suggestions and adopted the changes that improved clarity without making the presentation heavier. In particular, we clarified the use of $q$, the gradient parameter, the transition notation, and the relationship between deterministic policies and the induced stochastic action rule.
>
> Regarding the prior, we removed the statement about a “uniform prior” rather than expanding it. The  model section  explains that the paper works directly with the unnormalized posterior over policies. In this setting, a separate prior/conditional factorization would be artificial: the MDP and reward specification already induce the unnormalized target. This remains a Bayesian inference formulation, but the prior is not introduced as a separate modeling object.
>
> Regarding inverse temperature, we did not introduce $\beta$ as an additional algorithmic hyperparameter. This is intentional. One point of the paper is that, in the proposed formulation, the reward scale is part of the domain specification rather than a tunable algorithmic temperature, unlike in entropy-regularized methods such as SAC. To make the empirical comparison more symmetric, however, we added VSMC reward-scale sweeps for Blackjack, in addition to the SAC temperature comparison.
>
> Finally, we revised the experimental presentation. Blackjack now includes reward-scale results for VSMC. Academic Advising now includes CCDF plots. For Triangle Tireworld, we did not add CCDF plots because the returns are concentrated near the two dominant outcomes, making the CCDF comparatively uninformative. Instead, we added bar plots showing stuck and win probabilities for each instance. Thus, Blackjack, Triangle Tireworld, and Academic Advising now all present both tabular summaries and plots of the relevant distributional behavior.
>
> We hope these revisions address the main concerns while keeping the response proportionate to the current revision cycle.

---

> > ### Comment · Reviewer_QZSq · 2026-07-16
> >
> > I would like to thank the authors for their response. I have follow-up comments and questions.
> >
> > 1. I have to admit that additional examples and especially explanations on the difference after example (the last paragraph of Section 3) make it even more difficult to understand the value and any physical sense of the optimal policy $p^\star$, and the paper become substantially *less* readable after the revision. Also, I'd like to agree with Reviewer Fo96 that the paper misses actual motivation of introduction of this object except that it is distinct from MaxEnt RL one: any additional interpretation or "fixing" some failure modes of MaxEnt RL would work like that, but I do not see any of it in the current revision.
> >
> > 2. The new theoretical results are very confusing to me, and I stopped understanding the algorithm and what is this $q$-policy is at all (and the discussion paragraph in the end of Section 2 makes it even harder to understand). Is the policy $q$ Markovian or it is history-dependent in your algorithm? From point of view of Propositions 1 and 2, it seems that this policy is history-dependent, and, if my interpretation of Proposition 2 is correct, its marginals coincide with the optimal policy. But in this case, why do we even need to do any SMC if the proposal already assumed to be equal to the optimal policy up to marginalization? If the key case is when the proposal is not optimal (which is my current believe), then the paper still lacks the analysis (or, I'd say, a sanity-check) with a growing number of particles $N$.
> >
> > 3. I do not think that the discussion on the level "It is not the same thing" is enough given a striking similarity between your variational objective and the entropic risk-sensitive RL objective. I strongly believe that this similarity can be used to gather additional interpretations of the proposed optimal policy formulation. Additionally, it serves as an additional baseline which should be used as a comparison in the experiments if these formulation are indeed very distinct.
> >
> > Regarding the inverse temperature parameter, I respectfully disagree: in the MaxEnt RL, you can also fix a temperature equal to 1 and change only the reward scale and got the same behaviour. The dependence on the reward scaling is an expected property of any temperature-like quantity and not a novel feature of the approach.

---

> > > ### Author Response · Authors · 2026-07-16
> > > **Why do we need to do SMC**
> > >
> > > Variational SMC is a method of stochastic variational inference. The algorithm implements the loop of stochastic variational inference. $q_\theta$ is a variational family, it is parameterized by some parameter vector $\theta$.  At each step of the loop, VSMC computes the gradient of ELBO (evidence lower bound) by $\theta$, and update $\theta$ so that ELBO increases.
> > >
> > > VSMC is a _particle_ SVI algorithm, related to importance weighted autoencoders (IWAE - https://arxiv.org/abs/1509.00519), for example. The original VSMC paper establishes and analyzes the connection (Figure 1 and accompanying explanations on pages 1 and 2 - https://arxiv.org/abs/1705.11140). On every turn of the loop, VSMC draws a number ($N$) of particles from the proposal distribution. These particles are used to compute the ELBO.
> > >
> > > On problems with many time steps, such as MDP planning, a particle set without resampling  effectively collapses to a single particle. If one just samples several policies from $q$  independently and runs them through to completion (reach an absorbing state or hitting the maximum episode length), the weights of the particles will be very uneven, with one particle usually completely dominating the rest, thus resulting in computing the ELBO of VB (Variational Bayes). The IWAE paper (https://arxiv.org/abs/1509.00519) explains why IWAE is qualitatively different than VB and how multiple particles improve the ELBO. But with long trajectories IWAE degrades to VB.
> > >
> > > Variational SMC (not introduced in this work, VSMC is a work on which this submission is based, there is no connection between the authors of this paper and the authors of the VSMC paper) fixes this degradation by introducing SMC (particle filtering). SMC, through resampling, ensures that the particles used to compute the ELBO have similar weights, and the effective sample size of the particle set used to compute the ELBO is closer to the number of particles  than to 1.

---

> ### Author Response · Authors · 2026-07-16
> **Failure mode of MaxEntRL**
>
> MaxEntRL optimizes the sum of the expected reward and the entropy regularizer. As a result, MaxEntRL will encourage the agent to stay in a loop to increase the entropy regularization step. PolicyVSMC policies avoid loops. This is illustrated on the Grid World domain: a MaxEntRL agent deliberately bumps into the wall at the sides and corners of the grid, waiting for a stochastic _action_ (or _transition_) to take it to the next cell, thus increasing the entropy regularizer term at the cost of a small increase of the trajectory cost due to an extra step. VSMC policies are deterministic, the only way to leave a loop is through a stochastic _transition_, and actions that enter a loop are strongly discouraged. In the extreme case of a deterministic MDP with cycles, MaxEntRL will loop happily with high enough regularization; VSMC policies will almost never enter loops because a deterministic policy in a deterministic  MDP cannot leave a loop. This is **empirically illustrated**, visualized on policy maps and discussed (last paragraph of Section 6.1 and Figure 3)  on the gridworld domain. A similar behavior is also observed in the tire worlds and in the academic curriculum (due to self-cycles).
>
> That said, the main contribution of the paper is establishing that **stochastic actionable policies that balance between optimality and uncertainty can be obtained on theoretically sound grounds of the theory of probabilities when the magnitudes of rewards are interpreted as an expression of uncertainty about agent's preferences** rather than just  fixing MaxEntRL. MaxEntRL pathologies, **well known in MaxEntRL literature** cited in this paper and **empirically illustrated** in this submission, are artifacts of applying heuristics  (entropy regularization) to improve robustness  via stochastic policies, instead of building policy inference on sound  foundations of the theory of probability and  principles of statistical modeling.

---

> ### Author Response · Authors · 2026-07-16
> **Whether VSMC is over Markovian policies?**
>
> The policies drawn from $q$ are **Markovian** and **deterministic.** For a deterministic policy to be Markovian, it must select the action as a (deterministic) function of the state $a = \pi(s)$. When a deterministic Markovian policy is unrolled over time through sampling rather than sampled ahead of time for all states in the domains, $\pi(s)$ has to be memoized so that $\pi(s)$ returns the same $a$ for each $s$ visited. The history dependency in the algorithm is the incremental sampling of a Markovian deterministic policy, the policy in each particular step depends **ONLY** on the **SAME** policy in the **SAME** state.

---

> ### Author Response · Authors · 2026-07-16
> **Growing number of particles**
>
> Paper "Tighter variational bounds are not necessarily better" (https://arxiv.org/abs/1802.04537) cited in this submission analyses dependence of particle based stochastic variational inference (which is the tool used in this work for inference)
> on the number of particles. In particular, Figure 1 (page 4) and accompanying explanation analyse the dependency of the gradient estimates on the number of particles ($K$) grows, the mean and variance  of gradients decrease . Theorem 1 in that paper formalizes this phenomenon.
>
> Intuitively, the ELBO of VSMC (or IWAE) is the  **log sum of weights** of the particles in the particle set. The **sum of weights** of SMC is an _unbiased_ estimate of the evidence/marginal likelihood. This is a well known basic property of sequential Monte Carlo. Due to concavitiy of $\log(\cdot)$, the log sum of weights is _biased_ relative to the log marginal likelihood (on average, the log sum of weights of a particle sets will be lower than the log evidence, despite the sum of weights being equal on average to the evidence). This directs the optimization in the SVI loop, driving the parameters of the proposal distribution $q_\theta$ so that there is less variance in the evidence estimate.
>
> As the number of particles $N$ increases, the variance of the evidence estimate decreases, and the estimate approaches the true value as $N \to \infty$, which degrades ELBO gradients to zero. Therefore, VSMC (and policy VSMC in this paper by extension) is run with a small number of particles.
>
> Theoretically, two particles are sufficient. Practically, SMC does not resample well with 2 particles, leading to slower convergence for reasons _unrelated to the properties of VSMC ELBO estimate_. Original VSMC paper's experiments were run with 4 particles. In this paper we run with 10 particles. Running with 5 or 20 particles will slightly affect inference times (there will be domains with faster and domains with slower convergence). The inferred distributions will be the same, up to numerical approximation and the stochastic nature of the inference algorithm. We will be happy to report the experiments with 5 and 20 particles (we have the runs) but the inferred policies are virtually indistinguishable from already reported ones.

---

> ### Author Response · Authors · 2026-07-17
> **should entropic risk-sensitive RL be used in the experiment.**
>
> In the related work, we explained that entropic risk-sensitive RL (Eq. 56) is a variant of Levine's optimality variable RL (Eq. 57), where stochastic transitions are modeled as though the _transition distribution can be controlled by the agent_ (rather than sampled). This is a possible probabilistic model in general, but this model _does not reflect the stochastic process of an MDP agent_ going through the environment. Stochastic transitions are not latent variables, they cannot be inferred upon; that's the source of 'optimism bias' of  Levine's approach, and of the 'pessimism' bias of risk-sensitive RL.  These approaches do not solve MDPs, they solve different problems, which are not Markovian.  _This is what we argue about in the related work section._
>
> That said, we **do provide a comparison of our method to that of Levine's** for gridworlds, in figures  2b and 2c (dropping shared dynamics makes stochastic transitions inferrable as in the optimality-variable approach), and show the kind of pathology
> it causes in stochastic domains.
>
> We will also be happy to add experiments for risk-sensitive RL if the reviewers agree that this is a welcome addition to the submission (while still maintaining our opinion that these algorithms do not solve MDPs in the sense entropy-regularized RL or policy planning do: they solve a different kind of decision process, in which the agent controls stochastic transitions of the environment - in the optimality-variable approach, the agent _increases_ probabilities of state transitions (**not** of actions) with higher expected return, in the risk-sensitive approach the agent _decreases_ probabilities of such state transitions).
>
> 1. What **SOTA entropic risk-sensitive RL** algorithm should we compare our results against so that the comparison is fair?
> 2. Note that risk-sensitive RL algorithms, as published and implemented, .e.g. here https://github.com/kryptologyst/Risk-Sensitive-Reinforcement-Learning **do not infer a stochastic policy** with controllable entropy. They estimate a Q function that is then acted upon deterministically (arg max). Provided that published risk-sensitive RL algorithms emit deterministic policies and are analysed as such, how should we modify the risk-sensitive RL approach to obtain a  stochastic actionable policy for baseline/comparison?
> 3. Given that you recommend a version of an entropic risk-sensitive RL algorithm that does produce a stochastic policy with controllable entropy, **what should we control  for** in the experiments? For comparison of PolicyVSMC with SAC, we control for the mean return. With risk-sensitive RL algorithms, controlling for the mean return is not a meaningful comparison. Should we control for quantiles instead?  Which quantiles?

---

> ### Author Response · Authors · 2026-07-17
> **uncertainty about the transition model**
>
> Risk-sensitive RL implies (and provides a way to take care of) **uncertainty about the environment dynamics** --- the transition distribution $p(s'|s, a)$.  This is indeed an interesting topic, but _out of focus_ for this paper.  One can indeed author a paper comparing a probabilistic approach in which the policy model is as in this work, **but** transition probabilities are uncertain and a prior is given. Then, an inference algorithm (probably, but not necessarily VSMC-based) can be developed, and that algorithm  can and should be compared to a RL algorithm that infers stochastic policy with controlled entropy under risk-sensitivity objective.
>
> However, we strongly believe that including a discussion of the above in the current work would blur the presented result and make the work _less legible_ and overly long. While we are ready to extend the empirical session with new baselines, we urge the reviewers to reconsider this suggestion to keep the paper properly focused and coherent. **We revised the submission to clarify that risk-sensitive RL addresses a different kind of uncertainty and thus cannot serve as a baseline for experiments in this work.**

---

### Review · Reviewer_beUy · 2026-06-12

**Summary Of Contributions:**

This paper presents a novel framework for planning in Markov Decision Processes (MDPs). In contrast to trajectory-centric planning-as-inference methods, which are based on the randomness in the outcomes of trajectories, the authors propose a new Policy Inference formulation. In this framework, inference is performed over policies rather than trajectories. A posterior distribution of policies is maintained, which is defined by the expected returns of the policies. The induced control policy marginalizes over the policies in such a distribution. The authors demonstrate a method for performing inference using variational sequential Monte Carlo (VSMC), and they present a variety of empirical experiments to investigate the resulting behavior of Policy Inference methods.

**Audience:**

Yes

**Audience Explanation:**

To the reviewer's knowledge, policy inference frameworks, which explicitly consider the randomness over optimal policies for the MDP, are a relatively underexplored class of approaches towards decision-making under uncertainty. While policy inference itself may not be a completely novel framework, the proposed VSMC-based inference method points towards a practical approach of implementing such an agent and will be of interest to researchers and engineers investigating the proposed and similar MDP planning frameworks.

**Broader Impact Concerns:**

The reviewer believes the paper is compliant with all relevant considerations here.

**Claims And Evidence:**

Yes

**Claims Explanation:**

The authors provide mathematically sound proofs for claims made regarding the alignment between the VSMC target and the posterior-induced stochastic policy. Empirical results are supported with experimental data presented in clear and readable figures.

**Requested Changes:**

The reviewer stands in favor of acceptance, and the following points serve as recommendations for the authors in preparing their final draft.

1. Try to be more comprehensive and specific with regards to the motivation of the Policy Inference framework. Why would we, in practice, desire to acknowledge some level of uncertainty regarding the optimal policy? Is it due to partial observability, model uncertainty, or a desire to discover diverse modes of behavior (as in maximum entropy RL).

2. The authors are encouraged to be more specific about the particular gaps to be addressed by Policy Inference. For instance, in Section 4, the authors claim that "single-trajectory objectives are often underdispersed and prone to mode collapse." Further explanations and elaborations would be desirable here.

3. The authors are encouraged to provide more clarity on whether Policy Inference concerns only deterministic policies or stochastic policies as well. In Section 4 the authors claim that "the formulation also admits inference over stochastic policies." If the authors would like to maintain this claim, they might want to at least offer a sketch on why VSMC can be reliably generalized.

4. In the Experiments section, the authors are encouraged to explore how the differences in behavior between VSMC and SAC relate to the fundamental differences between the Policy Inference / entropy-regularized formulations.

---

> ### Author Response · Authors · 2026-06-16
>
> 1. On the motivation for representing uncertainty over the optimal policy
>
> The motivation is not primarily partial observability, model uncertainty, or a desire to discover diverse modes of behavior. Rather, the work is motivated by the view that reward magnitudes carry information, not only the ordering of policies. For example, randomizing between receiving 1 cent and 2 cents is qualitatively different from randomizing between receiving $1000$ and $2000$, even though the preference ordering is the same. The proposed formulation makes this intuition explicit: reward scale controls how strongly one policy is preferred over another, and Bayesian normalization turns these relative strengths into posterior uncertainty over deterministic policies. Thus uncertainty over the optimal policy reflects uncertainty or softness in the preference encoded by the reward scale, rather than exploration noise or entropy regularization. We revised the introduction to make this motivation explicit.
>
> 2. On underdispersion and mode collapse in single-trajectory variational objectives
>
> The sentence about single-trajectory objectives motivates the choice of VSMC as an inference tool. In this setting, a single sampled rollout gives a high-variance and local view of the policy posterior; optimizing such an objective can concentrate the proposal on one posterior mode and underestimate posterior dispersion. This is the standard mode-seeking/underdispersion behavior of common variational objectives for multimodal posteriors. We added citations to this literature at the relevant sentence. VSMC is used because its multi-particle objective provides a more robust particle approximation than a single-rollout structured variational objective.
>
> 3. On deterministic versus stochastic policies
>
> We reemphasized at the opening of the Inference section that the algorithmic part of the paper performs inference over deterministic policies. This is sufficient for stochastic control at execution time: stochasticity arises by marginalizing posterior uncertainty over deterministic policies, not by making each policy intrinsically stochastic. This is also why the VSMC construction in the paper is specialized to first-visit action assignments for deterministic policies.
>
> The probabilistic formulation itself can be extended to stochastic policy classes, and VSMC could indeed be adapted to such a setting by treating the stochastic-policy parameters or stochastic action rules as the latent policy object. We deliberately avoid developing this variant in the paper because it would add a second layer of action-level randomness and distract from the main point: posterior uncertainty over deterministic policies already induces a stochastic control rule. If the reviewer or editor believes this would be useful, we can add an appendix sketching how the VSMC construction extends to stochastic policies.
>
> 4. On connecting empirical behavior to the underlying formulations
>
> We agree that this connection is important. This is the main purpose of the Grid Worlds section: the domain is simple enough that the induced policies can be visualized directly, and the ablations isolate the algorithmic consequences of deterministic-policy inference. In particular, the VSMC/SAC comparison illustrates a concrete difference between the formulations: entropy-regularized SAC can assign probability to boundary-directed actions because they increase action entropy, whereas policy inference assigns posterior mass to deterministic policies, for which repeatedly choosing a boundary action is costly unless escaped through environment stochasticity. Thus the observed behavior follows from the distinction between action-level entropy regularization and posterior uncertainty over deterministic policies.
>
> We expanded the Discussion to make this connection explicit when summarizing the grid-world results.

---

### Review · Reviewer_Fo96 · 2026-06-16

**Summary Of Contributions:**

The main claim (repeated in abstract and intro several times) is that MDP planning can be naturally cast as Bayesian inference over deterministic policies, or, implicitly, that this view is useful for RL problems. Such inference can be performed by variational SMC in the policy space.

The paper provides an algorithm that is claimed to perform SMC over the space of policies by sampling actions in the course of a trajectory rollout. It is then applied to several RL problems to illustrate the properties of the marginal (posterior predictive) policy, in comparison to entropy-regularised RL.

I have doubts about the correctness of the algorithm. The claims about differences between the marginal policy and the max-ent policies are not clearly stated, and the ones that are made are not well-supported empirically. There are also concerns about the writing.

**Additional Comments:**

- In (1), is REWARD the same as $R$? Can absorption conditions currently in text after this equation be written out in maths (e.g., $R(s,a,s')=0$ if $s\in G$)?
- Description of VSMC could be clearer.
  - The spaces in which the variables lie is never stated.
  - It is not said after (3) that $q_\lambda$ is some parametric family of distributions (over which space?) that must have tractable sampling and tractable density, nor what it has to do with the production of particles and weights.
  - Minor: some instances of `,` instead of `\,` in (13).
  - I fear the whole §2.2 could be inaccessible to readers who are not already closely familiar with SMC. On the other hand, the proposition is a standard derivation, essentially the idea of hierarchical variational inference, and the space may better be used to explain how the SMC procedure actually works.
- In §3, there are some imprecisions.
  - Setup should be given earlier. What space is $\pi$ in? Are $S,A$ finite, so it is in a product of simplices? (I assume it must be so, since otherwise the space of policies is infinite-dimensional and we cannot make sense of the density.) Or is it the subspace of deterministic policies? Further, §3.1 does not discuss a prior over policies, but §3.2 talks about a uniform prior (which is implicit in the base density with respect to which the unnormalised density is defined).
    - Whether we consider deterministic or all policies affects the marginal policy, but it is not until §4 that it is stated that we consider deterministic policies.
    - Similarly, in §4 we find the answer to the countability/finiteness question.
  - In the first paragraph, the unbiased estimator should be of $\log\tilde\pi$, not of $\tilde\pi$.
- Proposition 2 speaks of an amortised approximation before it is introduced.
- Experiments:
  - It is hard to conclude from the experiments in §6.1 that use of the optimal stochastic policy is somehow preferable to the max-ent policy.
  - Presentation: What is the difference between the information shown in Table 1 and Figure 4? Any reason to have both? Aren't "stuck" and "goal" complementary in Figure 5?
  - In §6.3, SAC is only tested with $\alpha=1$, while VSMC is considered with tempered reward. In line with the other sections, SAC should also tested in the peaky reward setting. However, even in the default setting, SAC is achieving higher expected return and success probability than the basic VSMC.
- The importance of the horizon $H$ is not tested. Relatedly, can the results be generalised to the case of discounted reward?
- Does the first example (bandit) not generalise to any acyclic deterministic environment (with actions in bijection with successor states) with a fixed number of actions before termination?

**Audience:**

Yes

**Audience Explanation:**

Variants of entropic RL are relevant to the RL community itself, but also used in sampling/inference/discovery applications.

**Claims And Evidence:**

No

**Claims Explanation:**

The distinction between inference over policies and over trajectories (entropy-regularised RL) is repeated many times: in the abstract, several times in the introduction, just before §3.1, in §3.1, in §3.2, .... However:
- More discussion of the importance of the distinction, and the differences between the max-ent policy and the optimal stochastic policy under preference uncertainty (i.e., the marginal policy under the posterior this paper defines) would be informative -- there should be a clear statement of the claims.
- Some claims are made about these differences, hidden in the experiments section (e.g., heavier tails in §6.4). However, they are not well elucidated by the numerical results, as detailed below.
- Importantly, it is not explained why the difference are important/desirable (e.g., "empirical takeaways" in §7 just restates some observations from the experiments without any claim of generality).

**Requested Changes:**

- Critical: The concerns about claims and evidence above should be resolved. Some more questions and details of the concerns are mentioned in the comments below.
- Critical: The following questions should be answered. Can the marginal action distributions of the policy posterior be intrinsically characterised (by a local condition, similar to the soft Bellman equation optimised by entropy-regularised RL)? What is the difference between the proposed policy posterior and the logit quantal response equilibrium, where the environment is an opponent with a fixed strategy?
- Would strengthen the work: Writing concerns (both methodology and experiments) detailed in comments below should be addressed.
- Critical: A rollout of SMC only samples the deterministic actions for a policy at the states that are visited along the rollout, i.e., the SMC loop does not result in a sample from the space of deterministic policies! This seems to invalidate the arguments for correctness of the SMC. Notice that "partial policies" may capture differing amounts of prior mass because the sizes of the spaces of action assignments for the states unvisitable under those partial policies may differ. If this is incorrect, please explain where I go wrong.

---

> ### Author Response · Authors · 2026-06-16
>
> 1. On whether Policy-VSMC is SMC over the space of policies
>
> We would like to clarify that the paper does not claim that Policy-VSMC performs standard SMC over the full space of policies. VSMC is a variational inference method built around an SMC estimator, but it is not itself simply SMC targeting a posterior over complete policies. In our construction, each SMC sweep is run over simulated rollouts, with policy-action assignments sampled sequentially as they are encountered along a trajectory. The resulting SMC log-evidence estimate is then used as a variational objective for learning an amortized proposal over policy assignments. Thus, the role of the SMC sweep is to provide a multi-particle variational lower-bound estimator for inference over the policy posterior, not to produce exact SMC samples from the full policy space in a single sweep.
>
> 2. On the difference between the posterior marginal policy and maximum-entropy policies
>
> The difference between the posterior marginal policy and a maximum-entropy policy is primarily a difference in their defining principles, not an empirical hypothesis established only through experiments. The posterior-induced marginal policy is a property of the MDP together with the probabilistic model in Section 3: rewards define an unnormalized density over deterministic policies, and the stochastic action rule is obtained by marginalizing that posterior. By contrast, a maximum-entropy policy is the optimizer of a different control objective, namely expected return plus an entropy term. Thus the two policies are different objects even before any algorithm is introduced. The experiments are not intended to prove this conceptual distinction; rather, they illustrate that the distinction can lead to different observable behavior in concrete domains.
>
> 3. On the importance of the difference
>
> We agree that the importance of this difference can be emphasized more clearly. The paper is written within the broader framework of casting planning as probabilistic inference, a well-established line of work in which the benefits of probabilistic formulations are often taken as background: they provide a principled modeling language, make uncertainty explicit, and enable the use of a broad range of approximate inference tools. However, we recognize that this motivation should be made explicit rather than assumed. We have therefore revised the introduction to explain why it is useful to formulate planning as Bayesian inference over policies: the formulation preserves the expected-return semantics of the MDP while representing uncertainty over optimal behavior directly and making general-purpose inference methods applicable to planning.
>
> 4. On intrinsic characterization of the marginal action distribution
>
> The marginal action distribution is intrinsically characterized by the policy posterior defined in the probabilistic model. In particular, Eq. (15) assigns an unnormalized density to each deterministic policy via its expected return, and Eq. (17) defines the marginal action distribution by integrating this posterior over policies. The action probability at a state is therefore the posterior mass of deterministic policies that prescribe that action at that state. This differs from a soft Bellman equation because the distribution is over complete policies rather than directly over their statewise marginalizations; the statewise stochastic policy is obtained only after marginalizing the policy posterior. To make this connection explicit, we expanded Eq. (17) to show the marginalization over the policy posterior induced by Eq. (15).
>
> 5. On logit quantal response equilibrium
>
> There is a formal resemblance between our posterior and a logit response over pure strategies when an MDP is viewed as a game against an environment with a fixed strategy: deterministic policies play the role of the agent's pure strategies, the transition kernel fixes the environment's behavior, and J(pi) is the expected utility of pure strategy pi. In fact, the more direct connection to logit response is through entropy-regularized RL: in a one-step problem, entropy-regularized optimization yields the usual logit response, and in an MDP it yields a statewise softmax policy characterized by the soft Bellman equation. Our construction differs from this. We place the Boltzmann distribution over complete deterministic policies and only then marginalize to obtain statewise action probabilities. Thus, even under the fixed-environment game interpretation, the proposed object is a Boltzmann posterior over complete policies rather than the statewise logit-response policy associated with entropy-regularized RL.

---

> > ### Author Response · Authors · 2026-06-16
> >
> > 6. On partial policy assignments in an SMC sweep
> >
> > This concern is addressed by Proposition 1, "Posterior-conditional target of the VSMC proposal," which was added in the latest revision as background because it summarizes the standard variational interpretation of SMC rather than introducing a new result. The SMC sweep need not instantiate a complete policy in order to define a valid sequential variational approximation. As in standard VSMC, the proposal samples the next latent variable conditional on the variables already sampled; here, the sequential latent variables are the policy assignments encountered on first visits. Proposition 1 states that the optimal sequential proposal is the corresponding posterior conditional. Specializing this to the MDP setting gives Proposition 2: on a first visit to state s_t, the ideal proposal samples A_{s_t}=pi(s_t) from the posterior conditional given the previously fixed policy assignments. Unvisited state-action assignments are therefore marginalized rather than explicitly sampled. Their prior mass is accounted for by the posterior conditional over the partial assignment, exactly as in any sequential factorization of a posterior.
> >
> > This is also the usual logic of stochastic variational inference: a stochastic optimization step need not instantiate all latent variables in the model, only an unbiased or otherwise valid stochastic estimate of the variational objective or its gradient. VSMC can be viewed in this sense as a particle-based stochastic variational objective. A single SMC sweep samples only the latent variables needed for that stochastic estimate, while the optimized proposal represents the amortized posterior conditionals. Thus the fact that one rollout samples only the policy assignments relevant to that rollout does not invalidate the variational inference argument; it is precisely the conditional-posterior structure formalized by Propositions 1 and 2.
> >
> > 7. On the notation $\mathsc{Reward}$ and $R$
> >
> > Yes. $\mathsc{Reward}(s,a,s')$ is the simulator-interface notation for the reward function $R(s,a,s')$. We use $\mathsc{Step}$ and $\mathsc{Reward}$ in small caps to denote the procedural interface exposed by the MDP simulator, whereas $\mathcal{T}$ and $R$ denote the corresponding mathematical transition kernel and reward function. To remove any ambiguity, we have made this explicit in the text around Eq. (1).
> >
> > 8. On writing the absorbing-state convention mathematically
> >
> > Yes. The statement that goal states are absorbing implies $\mathcal{T}(s\mid s,a)=1$ and $\mathcal{T}(s'\mid s,a)=0$ for all $s'\neq s$, together with $R(s,a,s')=0$, whenever $s\in G$. We opted for the explicit absorbing-state formulation in prose because absorbing terminal states are standard in episodic MDPs and the surrounding text already states that the process remains in the absorbing state and accrues no further reward.
> >
> > 9. On the spaces of variables and the proposal in the VSMC background
> >
> > The reviewer writes that "the spaces in which the variables lie is never stated" and that "it is not said after (3) that $q_\lambda$ is some parametric family of distributions (over which space?) that must have tractable sampling and tractable density, nor what it has to do with the production of particles and weights." The general VSMC background section is meant to describe VSMC before specializing to the MDP setting. The latent variables $x_t$ and observations $y_t$ therefore need not be countable or finite; they live in general measurable spaces. The countable-state/finite-action assumptions introduced later are assumptions of the particular Policy-VSMC implementation used in the experiments, not assumptions of VSMC itself. To make the general setup explicit, we have revised the VSMC background to state that $x_t\in\mathcal X_t$ and $y_t\in\mathcal Y_t$, with $(\mathcal X_t,\mathcal Y_t)$ measurable spaces. We also clarified that $q_\lambda$ is a parameterized family of probability kernels on the latent spaces, used by SMC to sample particles sequentially and to evaluate the proposal densities or masses needed for importance weights.
> >
> > 10. On accessibility of the VSMC background
> >
> > The reviewer writes that "the whole Section 2.2 could be inaccessible to readers who are not already closely familiar with SMC." Section 2.2 is intended as VSMC background rather than as a full SMC tutorial, but we agree that a concise procedural reminder can make the section easier to follow. We have therefore updated the first paragraph of the VSMC background to state explicitly what an SMC sweep does: particles representing partial latent trajectories are extended by sampling from a proposal kernel, assigned incremental importance weights, and optionally resampled.

---

> > > ### Author Response · Authors · 2026-06-16
> > >
> > > 11. On the policy space, deterministic policies, and priors
> > >
> > > The reviewer asks what space $\pi$ belongs to, whether the state and action spaces are finite, whether the formulation is over stochastic or deterministic policies, and why Section 3.2 mentioned a uniform prior. The probabilistic model in Section 3 is intentionally stated at a level that does not commit to a particular representation of the state or action spaces. The object of inference is a policy, and the unnormalized density is defined over the policy class under consideration. In the deterministic-policy case used in the experiments, this is a class of maps $S\to A$; if stochastic policies are considered, the policy class is instead a class of conditional distributions or densities $p_\pi(a\mid s)$. Nothing in the probabilistic formulation requires $S$ and $A$ to be finite. Countability of states and finiteness of actions are introduced in Section 4 only for the particular VSMC implementation, where revisit memoization and categorical first-visit proposals are used. This separation is deliberate and is also reflected in the discussion of continuous-domain extensions.
> > >
> > > Regarding the prior, we removed the statement about a "uniform prior" rather than expanding it. The model section explains that the paper works directly with the unnormalized posterior over policies. In this setting, a separate prior/conditional factorization would be artificial: the MDP and reward specification already induce the unnormalized target. This remains a Bayesian inference formulation, but the prior is not introduced as a separate modeling object.
> > >
> > > 12. On the use of "amortized" in Proposition 2
> > >
> > > The reviewer notes that Proposition 2 uses "amortized" before it is introduced. We clarified the term at first use by adding that the learned state-indexed proposal is amortized in the sense of being shared across SMC sweeps and state visits.
> > >
> > > 13. On whether the experiments show that the posterior-induced policy is preferable to the maximum-entropy policy
> > >
> > > The experiments are not intended to show that the posterior-induced stochastic policy is uniformly preferable to a maximum-entropy policy in empirical return. The paper states this explicitly in the experimental setup and discussion. The point is conceptual: the posterior-induced policy is the control rule arising from the probabilistic model over policies, whereas a maximum-entropy policy is the optimizer of a different, entropy-regularized objective. The experiments are intended to illustrate that this difference in first principles can lead to different trajectory distributions and action-selection behavior, even when expected returns are close.
> > >
> > > 14. On potentially redundant tables and figures
> > >
> > > The Blackjack table and figure present the same outcome statistics in two complementary formats: the table gives numerical values with uncertainty, while the figure makes the comparison across methods visually immediate. The Triangle Tireworld outcome plots similarly visualize success and failure probabilities, even though in that domain the two outcomes are complementary. These additions were made in response to another reviewer's request for more distributional/visual presentation of the experimental results. We are happy to keep or remove redundant tables and figures according to the editor's and reviewers' preference.
> > >
> > > 15. On SAC temperature and VSMC reward scaling in Triangle Tireworld
> > >
> > > In Triangle Tireworld, SAC with the original rewards already performs well under the default setting $\alpha=1$. The additional VSMC result with $0.2\cdot r$ is included because the policy-posterior/VSMC machinery is sensitive to reward scale in this domain: with the original rewards, the posterior becomes very peaked and the particle-based optimization has high variance. Scaling the rewards down reduces this concentration and makes the induced behavior easier to compare to SAC.
> > >
> > > The goal of these experiments is not to compare algorithms by expected return or to tune both methods to their best possible performance. As stated in the paper, the goal is to compare the behaviors induced by different objectives when returns are in a comparable range. We are happy to run additional SAC experiments if the reviewer or editor believes a particular comparison is important; for example, SAC could be evaluated under several entropy weights $\alpha$ or under reward rescalings corresponding to the VSMC sweep. However, the interpretation would still be behavioral rather than a leaderboard comparison of expected return.

---

> > > > ### Author Response · Authors · 2026-06-16
> > > >
> > > > 16. On the horizon $H$ and discounted rewards
> > > >
> > > > The rollout horizon $H$ is an algorithmic truncation parameter. In benchmark domains derived from planning instances, we use the horizon specified by the problem setup; in the other domains, $H$ is chosen so that truncation does not materially affect the posterior-supported behavior observed in evaluation. The paper is not intended to study sensitivity to this algorithmic parameter, but rather the policy-inference formulation and the induced stochastic control rule.
> > > >
> > > > Discounted-reward problems can be handled within the same episodic framework. A discounted MDP can be represented as an equivalent episodic MDP by adding a transition to an absorbing terminal state with fixed probability $1-\gamma$ at each step, so that undiscounted return in the resulting episodic process corresponds to discounted return in the original problem. Thus the formulation specializes directly to discounted objectives.
> > > >
> > > > 17. On the scope of the bandit example
> > > >
> > > > No. The first worked example is a stochastic bandit, not a deterministic acyclic MDP. In the example, action $a_i$ has a Bernoulli reward: equivalently, it transitions to a reward-$1$ terminal outcome with probability $r_i$ and to a reward-$0$ terminal outcome with probability $1-r_i$. Thus the distinction illustrated there depends precisely on stochastic outcomes: the policy-posterior construction uses $\exp(\mathbb{E}[R_i])=\exp(r_i)$, whereas optimality-variable control-as-inference uses $\mathbb{E}[\exp(R_i)]=r_i e+(1-r_i)$. This separation would disappear in the deterministic one-step case.
> > > >
> > > > 18. On unbiased estimators of $\tilde p(\pi)$ and $\log \tilde p(\pi)$
> > > >
> > > > The reviewer notes that the stochastic estimator in the first paragraph of the probabilistic model should be an estimator of $\log \tilde p(\pi)$ rather than $\tilde p(\pi)$. We have clarified this point in the paper. The model section now distinguishes the two cases: posterior inference may use an unbiased stochastic estimator of $\tilde p(\pi)$, as in pseudo-marginal inference, or an unbiased stochastic estimator of $\log \tilde p(\pi)$, as in stochastic variational inference. In our rollout construction, Eq. (16) is an estimator of the unnormalized log probability $\log \tilde p(\pi)$.

---

> > > > > ### Comment · Reviewer_Fo96 · 2026-07-15
> > > > >
> > > > > Thank you for the responses. Here are my follow-up questions.
> > > > >
> > > > > In connection with (1) and (6), could you help me understand this example, with deterministic correspondence of transitions to actions?
> > > > > - There is a transition from s0 to s1, which is terminal and has reward r.
> > > > > - There is a transition from s0 to s2, which has zero reward.
> > > > > - There are transitions from s2 to states s21,s22,...,s2n, which each have zero reward.
> > > > >
> > > > > There are 2n deterministic policies in this environment (2 choices at s0, n choices at s2). Let us call p1,...,pn the policies that transition to s1 at s0 and q1,...,qn the policies that transition to s2 at s0, where the index denotes the action the policy would choose at s2. The expected rewards of the pj are r and of the qj are 0, so the posterior is uniform over the pj and the qj with the two groups having mass proportional to exp(r) and 1, respectively. The claim of Proposition 2 is that the optimal proposal at s0 would transition to s1 and s2 with probabilities proportional to those group weights, and this is what the learnt variational proposal should converge to. Is this correct?
> > > > >
> > > > > In (11) I do not understand the response. In continuous spaces, defining a distribution by its unnormalised density requires a choice of base measure, or prior. In the finite case, the base measure is the counting measure, which corresponds to a uniform prior. In the countably infinite case, if we take an improper prior (counting measure) and sample proportionally to exponentiated expected return, what guarantees the target is normalised and the posterior exists? There are similar questions in the stochastic finite-action case, where different choices of prior / base measure are possible: Lebesgue on pmfs, Lebesgue on logits modulo shift, Dirichlet, ...
> > > > >
> > > > > In (15) I continue to find odd in the different temperature parameters and would like to see analysis of the dependence on these parameters.
> > > > >
> > > > > Apologies, in (17) I meant an acyclic MDP with deterministic dependence of transitions on actions, so that a deterministic policy is equivalent to a choice of trajectory. The optimal max-ent RL policy  in this setting samples trajectories with mass proportional to exponentiated returns. Your analysis of the 1-step stochastic bandits seems to generalise to this case.
> > > > >
> > > > > Overall, some of the questions were addressed but I still fail to see the motivation for considering the posterior over policies in this way. The authors demonstrate that this gives a different marginal distribution over trajectories than max-ent objectives, but they also repeatedly claim that the goal is **not** to show this marginal distribution over trajectories is somehow more useful. What, then, is the point of this new formulation? We could say it is theoretical beauty and Bayesian principles, but the meaningful interpretations of the resulting object are lacking. (The response to (4) is unsatisfactory -- the new eq. 17 is essentially a tautology / the definition of the marginal, and it does not, for example, give a recursive condition that the marginal satisfies, such as the soft Bellman equation that the optimal max-ent RL policy respects.)
> > > > >
> > > > > Let me also note that I find the exposition in the paper and rebuttal even more difficult to follow than the original submission, and I venture a guess that some automated tools were used in the preparation (esp. given the very short time between the review and response). In particular, there is a large number of responses following the "not this, but rather that" schema, which detract from the substance. The paper and responses repeat time and again the point that the posterior over policies is distinct from other possible formulations (for example: in (4) I asked about an intrinsic characterisation of the policy posterior's marginal, and the authors responded by reminding that the new formulation is distinct from max-ent RL, a point already repeated many times in the paper).

---

> ### Author Response · Authors · 2026-07-15
> **Automation tools, example, sampling from unnormalized density, motivation.**
>
> **Automation tools:** I run pi (the coding agent, https://pi.dev/) with a frontier model on the case studies, the paper, and the reviews and rebuttals. I take full responsibility for the paper, the code, and the discussion here.  I wrote the Go code (VSMC and domains) exclusively by hand. I used codex, and then pi,  to adapt SAC implementation from CleanRL to discrete domains used here, and to translate domains implemented in Go  into Python for baselines, and to test that all of them define MDPs equivalent to the Go versions. The Julia code in the analysis/representation notebooks is hand written.  I used pi and a frontier model to edit and proof-read the paper. I read all of the cited papers, and stand by the citations included. I proved all propositions by hand and verified the proofs with the coding agent. I also verified the pseudocode in the appendix using the coding agent. I use the coding agent and a frontier model to verify that the paper contains all the details sufficient for re-implementation of the algorithm and the case studies on the grid word by directing the coding agent at the paper and asking to re-implement the algorithm (Policy VSMC) and grid worlds case studies, without access to the Go source code, in Python and JAX. This works. I am writing this particular response by hand.
>
>
>
> **Your example:** the unnormalized log probability of a policy starting with $s_0 \to s_1$ is $r$ . The unnormalized log probability of  a policy starting with $s_0 \to s_2$ is $0$. The unnormalized probabilites are $\exp r$ and $1$. _This holds for all three mentioned approaches_ (optimality variables, entropy regularized, policy inference). Expectation on rewards is taken when transitions are stochastic given an action for each given action. Marginalization on rewards (logsumexp) is taken when transitions are deterministic given an action, over all actions.
>
> **Unnormalized probabilities in continuous spaces**: one does not have to care about "base measure"/"prior" in continuous spaces to define a probabilistic model via an unnormalized probality density of the variable of interest. For example $p(x) \propto \exp (- x^2)$ is a fully specified probabilistic model with respect to the variable of interest $x$. To infer the distribution of $x$, one uses an inference algorithm, such as Markov chain Monte Carlo (random-walk [Metropolis-Hastings](https://en.wikipedia.org/wiki/Metropolis%E2%80%93Hastings_algorithm) being the simplest variant) or [rejection sampling](https://en.wikipedia.org/wiki/Rejection_sampling). If it weren't the case one wouldn't be able to construct [Bayesian statistical models](https://sites.stat.columbia.edu/gelman/book/BDA3.pdf).
>
> **Experiments with different temperatures**: I would be happy to run additional experiments for different temperatures, reflect the results in the tables and the plots, and discuss them. For which temperatures should I run  the additional experiments?
>
> **Equation (17) and the response on intrinsic characterization**: the response clarifies, and the equation formalizes, that in the policy obtained through the policy inference described in this submission, action selection is based on the global context rather than on local summary statistics, and thus cannot be represented similarly to soft Bellman backup as in entropy-regularized RL. It is not uncommon for a distribution to not have a summary statistics.
>
> **Motivation of the research**: neither the submission nor the rebuttal say  that the proposed approach is not 'better' than existing ones. What they say is that the theoretical analysis and the empirical evaluation do not mean to compare expected returns of the inferred policies using either the proposed approach or the baseline; on the opposite, the experiments are _controlled_ (e.g. by choosing SAC regularization) so that the expected returns are similar, and compare characteristics of inferred policies which are not captured by the the expected return. The message that the submission conveys is that **stochastic actionable policies that balance between optimality and uncertainty can be obtained on theoretically sound grounds of the theory of probabilities when the magnitudes of rewards are interpreted as an expression of uncertainty about agent's preferences**. Case studies illustrate qualitative differences between policies inferred via the proposed probabilistic model and via entropy-regularized RL.

---

> ### Author Response · Authors · 2026-07-18
> **Unnormalized densities are all we need for inference in continuous variable models**
>
> Take a look at this section from Stan documentation: https://mc-stan.org/docs/2_39/stan-users-guide/custom-probability.html
>
> It shows how to fully specify a model using unnormalized probabilities. Stan is a SOTA Bayesian modeling tool and a reference for established approaches, but this is not somehow "unique" to Stan. **Any statistical inference engine** works with unnormalized densities, that's what inference is (and that's the kind of inference VSMC performs in this submission).
>
> In particular, here is a relevant paragraph from that section. It explains that Stan **drops** constant normalization terms automatically because they are not needed for inference.
>
> **Unnormalized (log) densities are all we need for inference in continuous variable models.**
>
> ------
>
> Exponential distribution
>
> If Stan didn’t happen to include the exponential distribution, it could be coded directly using the following assignment statement, where lambda is the inverse scale and y the sampled variate.
>
> target += log(lambda) - y * lambda;
>
> This encoding will work for any lambda and y; they can be parameters, data, or one of each, or even local variables.
>
> The assignment statement in the previous paragraph generates C++ code that is similar to that generated by the following distribution statement.
>
> y ~ exponential(lambda);
>
> There are two notable differences. First, the distribution statement will check the inputs to make sure both lambda is positive and y is non-negative (which includes checking that neither is the special not-a-number value).
>
> The second difference is that if lambda is not a parameter, transformed parameter, or local model variable, the distribution statement is clever enough to drop the log(lambda) term. This results in the same posterior because Stan only needs the log probability up to an additive constant. If lambda and y are both constants, the distribution statement will drop both terms (but still check for out-of-domain errors on the inputs).